

# Compositional Evolution of Particle Phase Reaction Products and Water in the Heterogeneous OH Oxidation of Aqueous Organic Droplets

Man Mei Chim[1], Chiu Tung Cheng[1, †], James F. Davies[2], Thomas Berkemeier[3], Manabu Shiraiwa[4], Andreas Zuend[5], Man Nin Chan[1,6]

[1]Earth System Science Programme, Faculty of Science, The Chinese University of Hong Kong, Hong Kong, CHINA
[2]Chemical Sciences Division, Lawrence Berkeley National Laboratory, Berkeley, USA
[3]School of Chemical & Biomolecular Engineering, Georgia Institute of Technology, Atlanta, Georgia, USA
[4]Department of Chemistry, School of Physical Sciences, University of California, Irvine, Irvine, USA
[5]Department of Atmospheric and Oceanic Sciences, McGill University, Montreal, Québec, CANADA
[6]The Institute of Environment, Energy and Sustainability, The Chinese University of Hong Kong, Hong Kong, CHINA
[†]Present address: Atmosphere and Ocean Research Institute, The University of Tokyo, Tokyo, JAPAN

*Correspondence to*: Man Nin Chan (mnchan@cuhk.edu.hk)

**Abstract.** Organic compounds present at/near the surface of aqueous droplets can be efficiently oxidized by gas-phase OH radicals, which alter the molecular distribution of the reaction products within the droplet. A change in aerosol composition affects the hygroscopicity and leads to a concomitant response in the equilibrium amount of particle phase water. The variation in the aerosol water content affects the aerosol size and physicochemical properties, which in turn governs the oxidation kinetics and chemistry. To attain better knowledge of the compositional evolution of aqueous organic droplets during oxidation, this work investigates the heterogeneous OH radical initiated oxidation of aqueous methylsuccinic acid ($C_5H_8O_4$) droplets, a model compound for small branched dicarboxylic acids found in atmospheric aerosols, at a high relative humidity of 85 % through experimental and modelling approaches. Aerosol mass spectra measured by a soft atmospheric pressure ionization source (Direct Analysis in Real Time, DART) coupled with a high-resolution mass spectrometer reveal two major products: a five carbon atom ($C_5$) hydroxyl functionalization product ($C_5H_8O_5$) and a $C_4$ fragmentation product ($C_4H_6O_3$). These two products likely originate from the formation and subsequent reactions (intermolecular hydrogen abstraction and carbon-carbon bond scission) of tertiary alkoxy radicals resulting from the OH-abstraction occurring at the methyl-substituted carbon site. Based on the identification of the reaction products, a kinetic model of oxidation (a two-product model) coupled with the Aerosol Inorganic-Organic Mixtures Functional groups Activity Coefficients (AIOMFAC) model is built to simulate the size and compositional changes of aqueous methylsuccinic acid droplets during oxidation. Model results show that at the maximum OH exposure, the droplets become slightly more hygroscopic after oxidation, as the mass fraction of water predicted to increase from 0.362 to 0.424; however, the diameter of the droplets decreases by 6.1 %. This can be attributed to the formation of volatile fragmentation products that partition to the gas phase, leading to a net loss of organic species and associated particle phase water, and thus a smaller droplet size. Overall, fragmentation and volatilization processes play a larger role than the functionalization process in determining the evolution of aerosol water content and droplet size at high oxidation stages.

40



# 1 Introduction

Atmospheric organic aerosols can be continuously oxidized by gas-phase oxidants such as hydroxyl (OH) radicals and ozone ($O_3$) throughout their lifetime, with a chemical lifetime of several days to weeks (*Rudich et al., 2007; Jimenez et al., 2009; George and Abbatt, 2010; Kroll et al., 2015*). This radical-initiated heterogeneous oxidation of organic aerosols is an important aging process that can significantly change the aerosol composition and, therefore, alters the properties of aerosols, such as their light scattering ability, hygroscopicity, and cloud condensation nuclei (CCN) activity (*Broekhuizen et al., 2004; Shilling et al., 2007; Cappa et al., 2011; Wong et al., 2011; Lambe et al., 2011; Dennis-Smither et al., 2012*). For example, the heterogeneous OH oxidation of insoluble or sparingly soluble organic compounds (e.g. squalane and bis(2-ethylhexyl)sebacate) produces water-soluble reaction products that enhance the CCN activity of corresponding aerosols (*George et al., 2009; Harmon et al., 2013*).

Organic compounds dissolved within liquid aqueous droplets have found to be oxidized by gas-phase OH radicals at a much faster rate than within solid and amorphous aerosol particles (*Chan et al., 2014; Slade and Knopf, 2014; Davies and Wilson, 2015; Arangio et al., 2015; Slade et al., 2017*). This is attributed to shorter mixing times in liquid submicron-sized droplets, allowing the condensed phase material to diffuse rapidly within the droplet bulk and access the surface on the timescale of a reaction (*Shiraiwa et al., 2011; Berkemeier et al., 2013; Houle et al., 2015; Berkemeier et al; 2016*). Heterogeneous OH reaction rates of aqueous organic droplets have been found to increase with increasing relative humidity (RH) (*Slade and Knopf, 2014; Davies and Wilson, 2015*). This occurs as droplets become more dilute and less viscous with increasing water content in equilibrium with an increase in the ambient RH, allowing more rapid diffusion of reacting species from the bulk to the near-surface region where oxidation takes place effectively, leading to a higher overall oxidation rate. In our recent work, Chim et al. (2017) show that for the heterogeneous OH oxidation of aqueous 2-methylglutaric acid droplets, the same reaction products are formed over a range of RH (33.8–82.5 %). At a given reaction extent, the particle composition does not strongly depend on the RH in that case. Those results suggest that the particle phase water content does not alter reaction mechanisms significantly over the experimental RH. The RH and corresponding particle phase water likely influence the heterogeneous OH reactivity by their effect on viscosity and molecular diffusion.

Depending on the particle composition and environmental conditions, such as RH and temperature, particle phase water can be an important component of aqueous organic droplets (*Peng et al., 2001; Brooks et al., 2002; Braban et al., 2003; Marcolli et al., 2004; Mochida and Kwamura, 2004; Parsons et al., 2004; Chan et al., 2005, 2008; Mikhailov et al., 2009; Ma et al., 2013; Ganbavale et al., 2014; Chen et al., 2015; Jing et al., 2016*). Upon oxidation, the amount of particle-phase water can continuously vary in response to changes in particle composition. For example, heterogeneous oxidation can lead to the addition of new functional groups (e.g. hydroxyls and carbonyls) onto the parent organic molecules through functionalization processes (*Russell, 1957; Bennett and Summers, 1974*). The formation of oxygenated functionalization



products of increased water solubility can enhance the hygroscopic properties of the aerosols at a certain RH (*Kroll et al., 2015*). Alternatively, through a fragmentation process, organic molecules can decompose to form smaller products upon oxidation with fewer carbons than their parent molecules. The fragmentation products may show enhanced partitioning to the gas phase or remain in the liquid particle phase, depending on their effective volatilities. As with the functionalization

5   products, an increased proportion of oxygenated fragmentation products remaining in the particle phase can enhance the aerosol hygroscopicity. However, if fragmentation products partition preferentially to the gas phase, the aerosol organic mass and the amount of particle-phase water may decrease, depending on the extent of the volatilization. Overall, the competition between functionalization and fragmentation, and their interplay with volatilization, are crucial in the evolution of particle-phase reaction products and water within aqueous organic droplets during oxidation.

While heterogeneous oxidation generally leads to more soluble and oxygenated products, the hygroscopicity of organic particles should be correlated with the degree of average organic aerosol oxygenation, which can be characterized by bulk elemental composition such as the average oxygen-to-carbon (O/C) ratio and the average carbon oxidation state $(\overline{OS_c})$ (*Kroll et al., 2011*). However, hygroscopicity is not perfectly correlated with O/C ratio as different chemical functionalities and

15   molecular orientations influence the interaction with water and other compounds (*Rickards et al., 2013*). Thus, aerosols with the same average oxidation state and elemental composition do not necessarily exhibit the same hygroscopicity, as they may have a different molecular distribution of reaction products. In addition to the bulk elemental composition, knowledge of molecular composition of the particles before and after oxidation is needed to better understand how the aerosol hygroscopicity evolves during oxidation.

Recently, atmospheric pressure ionization techniques such as extractive electrospray ionization (EESI) (*Doezema et al., 2012*; *Gallimore and Kalberer, 2013*), Direct Analysis in Real Time (DART) (*Nah et al., 2013; Chan et al., 2013, 2014; Cheng et al., 2015, 2016; Zhao et al., 2017*) and Aerosol Flowing Atmospheric-Pressure Afterglow (AeroFAPA) (*Brüggemann et al., 2015*) coupled with high resolution mass spectrometers have been used to characterize the composition

25   of organic aerosols at a molecular level in real-time. These techniques are of particular interest for investigating the chemical transformation of aqueous organic droplets through heterogeneous oxidation because the composition of the droplets can be directly analyzed in their native states. With improved knowledge of the composition of the reaction products, the aerosol water content of complex organic mixtures can be reasonably predicted using aerosol thermodynamic models, several of which are also available online, such as the Extended Aerosol Inorganics Model, E-AIM, (*Clegg et al., 2001; Wexler and*

30   *Clegg, 2002*), UManSysProp v1.0 (*Topping et al., 2016*), and the Aerosol Inorganic-Organic Mixtures Functional groups Activity Coefficients (AIOMFAC) model (*Zuend et al., 2008; Zuend et al., 2011*). Coupling an oxidation kinetic model with an aerosol thermodynamic model can be a valuable tool to investigate how the composition of aqueous organic droplets (including water) changes during heterogeneous oxidation.



To attain more insight into how particle-phase reaction products and water content evolve upon oxidation, experiments were conducted to investigate the chemical evolution of molecular composition of aqueous methylsuccinic acid droplets upon heterogeneous OH oxidation at 85 % RH using an atmospheric pressure aerosol flow tube reactor coupled with the DART mass spectrometry. Methylsuccinic acid is one of the most abundant branched dicarboxylic acids observed in atmospheric

aerosols (*Li et al., 2015; Kundu et al., 2016*) and is chosen as a model compound to gain a more fundamental understanding of the heterogeneous OH chemistry of methyl substituted dicarboxylic acids (**Table 1**). Based on the identification of reaction products and proposed reaction mechanisms, an oxidation kinetic model (a two-product model) coupled with an aerosol thermodynamic model (AIOMFAC) is proposed to simulate the composition of aqueous methylsuccinic acid droplets during heterogeneous oxidation with variable OH radical exposure. The primary goal of this model is to examine how the

competition between functionalization and fragmentation reactions, in conjunction with volatilization, determines the evolution of the particle-phase products and water content during oxidation.

## 2 Experimental Approach

### 2.1 Heterogeneous Oxidation

The heterogeneous OH oxidation of methylsuccinic aerosols was carried out in an atmospheric pressure aerosol flow-tube

reactor. Experimental details have been described elsewhere (*Chan et al., 2014*). In brief, the aerosol was first generated by atomizing an aqueous methylsuccinic acid solution using a constant output atomizer (TSI Inc. Model 3076). A portion of the aerosol stream was diluted into a mixture of nitrogen, oxygen, ozone, and hexane and conditioned to a controlled RH of about 85 % at a temperature of 293 K before entering the reactor. Inside the aerosol flow-tube reactor, the oxidation of methylsuccinic acid aerosols was initiated by gas-phase OH radicals generated by the photolysis of ozone under UV

illumination at 254 nm. The OH concentration within the reactor was varied by changing the ozone concentration and quantified by measuring the decay of hexane, a gas-phase OH tracer, using gas chromatography coupled with a flame ionization detector (GC-FID; *Smith et al., 2009*). The OH exposure was ranged from 0 to $1.47 \times 10^{12}$ molecules cm$^{-3}$ s and the aerosol residence time was 1.3 min. Upon exiting the reactor, the aerosol stream was passed through an annular Carulite catalyst denuder and an activated charcoal denuder to remove ozone and other gas-phase species from the aerosol stream,

respectively. The size distribution of the aerosol leaving the reactor was measured using a scanning mobility particle sizer (SMPS, TSI).

The remaining flow was directed into a stainless steel tube heater, where the aerosol particles were fully vaporized at a temperature of 250 °C in order to measure the bulk composition. The gas-phase species were then directed into the

atmospheric pressure ionization region for real-time chemical characterization using the high resolution mass spectrometer (ThermoFisher, Q Exactive Orbitrap). A negative ionization mode, with helium as an ionizing gas, was used for the operation of DART ionization source (IonSense: DART SVP). In the ionization region, the acidic proton of the carboxylic



acid group in methylsuccinic acid and its reaction products is likely abstracted by the anionic oxygen ions ($O_2^-$) to generate the deprotonated molecular ions, $[M-H]^-$, which were subsequently sampled by the high resolution mass spectrometer (*Cody et al., 2005; Cody, 2008; Nah et al., 2013; Chan et al., 2013, 2014; Cheng et al., 2015, 2016; Chim et al., 2017*). Mass spectra were collected at 1 s intervals over a scan range from mass-to-charge (*m/z*) ratios 70–500, with each spectrum averaged over a 5-min sampling time with a mass resolution of 140,000. Mass calibration was carried out with standard solutions prior to experiments and the mass spectra were analyzed using Xcalibur (Xcalibur Software, Inc., Herndon, VA, USA).

## 2.2 Hygroscopicity Measurements

To determine the aerosol physical state prior to oxidation and validate the composition of the aerosol predicted by the AIOMFAC model, the hygroscopic behavior of methylsuccinic acid aerosols was measured using an aerosol optical tweezers setup (Biral AOT 100) (*Davies and Wilson, 2016*). An individual droplet (~10 μm in diameter) was isolated and held in an optical trap, in which it was exposed to a controlled humidity profile ranging from about 25 % to 87 % RH at a temperature of 295 K. The droplet size and refractive index, with a measurement precision of approximately 1 nm and 0.05 % respectively, were monitored using the wavelength position of cavity enhanced resonances in the Raman spectrum and the sizing algorithms of Preston and Reid (2013). An offline calibration was performed to correlate the mass fraction of solute (*mfs*) (here the organic component is the solute and water is the solvent) to the refractive index using bulk solutions and a digital refractometer (Atago PAL-RI), along with the refractive index value determined for the liquid-state pure solute component from the work by Marsh et al. (2017). Using the dispersion parameters from the sizing algorithms (*Preston and Reid, 2013*), the refractive index determined in the AOT was corrected to the wavelength of calibration and the *mfs* was determined as a function of RH for each measured value of refractive index.

## 3 Results and Discussions

### 3.1 Hygroscopicity of methylsuccinic acid aerosols

**Figure 1** shows the experimentally determined hygroscopicity of methylsuccinic acid, expressed as the mass fraction of solute (*mfs*) as a function of RH. In these equilibrium measurements, the RH is equivalent to the particle-phase water activity. Methylsuccinic acid droplets absorb or desorb water in a reversible manner in response to the set environmental RH. Since the droplets maintained a spherical shape over the entire experimental RH range, crystallization did not occur. As discussed in Sect. 2.1, during the aerosol flow-tube reactor experiments, aqueous methylsuccinic acid aerosols were always exposed to a sufficiently high RH and, thus, remained in a liquid state prior to oxidation.

For comparison, the hygroscopicity of methylsuccinic acid was computed using the AIOMFAC model. The model predictions show excellent agreement with the experimental data, especially at RH > 60 %, with an increasing degree of





deviation toward lower RH values (remaining within experimental uncertainty). For instance, at 85 % RH, the *mfs* interpolated from the experimental data and model value differs by 0.01 and at 30 % RH, the experiments suggest a *mfs* of $0.96 \pm 0.05$ versus a model prediction of 0.93. Given the generally good agreement between the experimental data and model predictions in this study, the use of the AIOMFAC model is considered to be a reliable approach to predict the

hygroscopicity of methylsuccinic acid aerosols before and after oxidation.

Recent measurements by Marsh et al. (2017) using the comparative kinetics technique applied in an electrodynamic balance (CK-EDB) also report hygroscopicity data for methylsuccinic acid (see **Fig. 1**). In contrast to the good agreement between experimental data and model predictions in this study, the CK-EDB measurements seems to slightly overpredict the

hygroscopicity (i.e. smaller *mfs* at a given RH) compared to both the AIOMFAC model and the AOT experimental data, even though the experimental data mostly agree within the stated uncertainties. The CK-EDB method would observe a smaller *mfs* at a given water activity if subtle particle-phase diffusion limitations slowed the evaporation rate of water from droplets to an extent not considered in the mass flux framework used by Marsh et al. (2017), in which homogeneous mixing of organic solute and water is assumed in the droplet bulk during net evaporation conditions. If such effects were to play a

role, at a given droplet size and *mfs*, the evaporation rate would tend to be slower and the data evaluation by the mass flux equation would predict a smaller droplet water activity. However, we note that at this point it remains unclear whether the CK-EDB experiments and data evaluation were affected by such mixing and mass transfer effects in the case of methylsuccinic acid droplet measurements. One other potential source of uncertainty is in the treatment of the density, which is required to relate the droplet radii measured in the CK-EDB to a mass fraction. However, given the uncertainties

associated with both the CK-EDB data of Marsh et al. (2017) and the AOT data reported here, there is a generally good agreement between the experimental and modelled hygroscopic data. Experimental data obtained from the hygroscopicity measurement is given in **Table S1** (Supporting Information).

## 3.2 Aerosol Mass Spectra

**Figure 2** shows the aerosol mass spectra of methylsuccinic acid before and after oxidation. Before oxidation, the single

dominant peak is the deprotonated molecular ion of methylsuccinic acid ($C_5H_7O_4^-$, *m/z* = 131). After oxidation (at the maximum OH exposure of $1.47 \times 10^{12}$ molecule $cm^{-3}$ s), two major peaks, one $C_4$ fragmentation product $C_4H_5O_3^-$ (*m/z* = 101) and one $C_5$ functionalization product $C_5H_7O_5^-$ (*m/z* = 147), are observed in addition to the remaining, unreacted methylsuccinic acid (see **Table 2**). A number of minor product peaks are also detected in the aerosol mass spectra; with each contributing less than 2 % of the total ion signal (see Supporting Information, **Table S2**). The chemical evolution of

methylsuccinic acid and the major reaction products as a function of OH exposure are plotted in **Fig. 3**. At the maximum OH exposure, as shown in **Table 2**, the $C_5$ functionalization product ($C_5H_8O_5$) is the most abundant species (48.4 %), followed by unreacted methylsuccinic acid (30.3 %), and the $C_4$ fragmentation product ($C_4H_6O_3$, 12.8 %). The sum of all minor products contributes to 8.5 % of the total products.



Since the standards of ionization efficiency of methylsuccinic acid and the reaction products, which are required for quantifying the aerosol composition from the DART mass spectra, are not available, same ionization efficiency is assumed for the parent compound and the reaction products. In addition, the reactions between peroxy radicals and/or hydroperoxy radicals can possibly yield organic peroxides and oligomers, which may thermally decompose in the hot DART ionization

source. These potential products may not be efficiently detected in the aerosol mass spectra; however, there was no indication of any fragment ions expected to be formed from the thermal decomposition of such products in the mass spectra (which would be detectable).

### 3.3 Heterogeneous Oxidation Kinetics

In order to quantify the kinetics, the intensity of the parent compound peak ($I$) after reaction is normalized by that before

reaction ($I_0$). As shown in **Fig. 4**, the decay of methylsuccinic acid due to oxidation by OH exhibits an exponential behavior and can be fit with an exponential function to obtain a bimolecular rate constant ($k$):

$$ln \frac{I}{I_0} = -k[\text{OH}]t \, , \tag{1}$$

where [OH] is the concentration of gas-phase OH radicals and $t$ is the reaction time. The fitted value of $k$ is estimated to be $(1.07 \pm 0.09) \times 10^{-12} \, \text{cm}^3 \, \text{molecule}^{-1} \, \text{s}^{-1}$, which can be used to compute the effective OH uptake coefficient, $\gamma_{\text{eff}}$, defined as

the fraction of OH collisions that yield a reaction (*Smith et al., 2009*):

$$\gamma_{\text{eff}} = \frac{2 \, D_0 \, \rho \, mfs \, N_A}{3 \, M \, \overline{c_{OH}}} \, k \, , \tag{2}$$

where $D_0$, $mfs$, and $\rho$ are the mean surface-weighted diameter, the mass fraction of solute, and the density of the aqueous droplets before oxidation, respectively. $M$ is the molar mass of methylsuccinic acid, $N_A$ is Avogadro's number, and $\overline{c_{OH}}$ is the mean velocity of gas-phase OH radicals. Prior to oxidation, the droplet diameter was determined to be $D_0 \sim 237.2$ nm. The

initial composition of the droplets (i.e. $mfs$) is determined from the hygroscopicity data depicted in **Fig. 1**. The density ($\rho$) is estimated using an additivity rule with the known pure component densities and mass fractions of water and methylsuccinic acid (1.4779 g cm$^{-3}$) at a temperature of 20 $^{\circ}$C. A value of $\gamma_{\text{eff}} = 1.02 \pm 0.17$ was determined with these parameters from Eq. (2). A value of $\gamma_{\text{eff}}$ slightly greater than one may indicate that secondary chemistry is occurring, which is discussed in the following Sect. 3.4.

### 3.4 Reaction Mechanisms

Plausible reaction mechanisms are proposed to explain the formation of reaction products detected in the aerosol mass spectra based on well-known particle-phase reactions previously reported in the literature (*George and Abbatt, 2010*). As shown in **Scheme 1**, the OH oxidation with methylsuccinic acid can be initiated by the abstraction of a hydrogen atom located on three different sites: tertiary backbone carbon site (**Path A**), secondary backbone carbon site (**Path B**), and the

primary carbon site of the branched methyl group (**Path C**). Depending on the initial OH reaction site, a variety of functionalization and fragmentation products can be formed. The detailed reaction mechanisms are described as follows.



### 3.4.1 Functionalization products

In the first OH oxidation step, one of the hydrogen atoms of the methylsuccinic acid can be abstracted by an OH radical. An alkyl radical is formed that reacts quickly with an $O_2$ molecule to form a peroxy radical. The self-reaction of two peroxy radicals can yield the major $C_5$ alcohol ($C_5H_8O_5$) and minor $C_5$ ketone ($C_5H_6O_5$) products through the Russell mechanism

(**R1**) and/or Bennett-Summers reactions (**R2**). Two alkoxy radicals can also be formed from the self-reaction of two peroxy radicals. They can then abstract hydrogen atoms from neighboring molecules (**R3**) to form $C_5$ alcohol products or react with $O_2$ molecules (**R4**) to form $C_5$ ketone products. It is noted that the alkoxy radicals may undergo isomerization. For instance, the isomerization of the alkoxy radical (**Path A, Scheme 1**) can yield the functionalization product ($C_5H_8O_6$), which is a minor product detected in the mass spectra. This may suggest that this isomerization might not be significant for the

formation of major products. The detail reaction pathways will not be discussed in Scheme 1. When the hydrogen abstraction occurs at the tertiary carbon site (**Scheme 1, Path A**), only an alcohol group can be added to the tertiary carbon site and the $C_5$ alcohol product is the only possible product. Depending on the initial OH reaction site, it is possible to form structural isomers of these alcohol and ketone products (**Scheme 1, Path B** and **Path C**).

### 3.4.2 Fragmentation products

The fragmentation products likely originate from the decomposition of alkoxy radicals (**R5**). For example, when the oxidation occurs at the tertiary carbon site (**Scheme 1, Path A**), the decomposition of the tertiary alkoxy radical yields the major $C_4$ ($C_4H_6O_3$) and minor $C_3$ ($C_3H_4O_3$) fragmentation products. An isobaric compound of the major $C_4$ fragmentation product can also be formed from the decomposition of the alkoxy radical formed at the secondary carbon backbone site (**Scheme 1, Path B**). $C_3$ fragmentation products generated from the OH abstraction occurring at the primary carbon site

(**Scheme 1, Path C**) are also detected in small amounts (< 0.1 %).

Alkoxy radicals can fragment into two smaller products via carbon-carbon scission. From the aerosol speciation data, the decomposition of an alkoxy radical results in a higher abundance of larger products than smaller products. For example, the decomposition of a $C_5$ alkoxy radical via Path A and Path B (**Scheme 1**) yields the major $C_4$ ($C_4H_6O_3$) fragmentation product

in a high relative abundance of 12.8 %, while the minor $C_3$ ($C_3H_4O_3$) fragmentation product is detected at 1.1 % abundance, and the $C_2$ ($C_2H_2O_3$) fragmentation product is not detected. The low abundance of smaller products is presumably due to their higher volatility and/or low yields as predicted using a structure-activity relationships (SAR) developed to estimate rate constants for the decomposition of gas-phase alkoxy radicals, based on the number and identity of the functional groups located near the carbon atoms at the decomposition site (*Peeters et al., 2004; Vereecken et al., 2009*). Using the tertiary

carbon site (**Scheme 1, Path A**) as an illustration, the absolute (formation) rate coefficient ($k_{SAR}$) of the $C_4$ product ($C_4H_6O_3$) is $2.06 \times 10^{12}$ s$^{-1}$, which is several orders of magnitude higher than that of the $C_3$ product ($C_3H_4O_3$) ($8.39 \times 10^5$ s$^{-1}$).



Overall, the major products detected in the aerosol mass spectra are likely 1$^{st}$ generation products and their formation pathways can be well explained by the proposed reaction scheme (**Scheme 1**). The 1$^{st}$ generation products can be further oxidized by OH radicals to generate 2$^{nd}$ or higher generation products via similar reaction mechanisms. Since the 1$^{st}$ generation products are the dominant species and minor products only account for less than 10 % of the total products, for simplicity, the formation pathways of the 2$^{nd}$ or higher generation products are not discussed here in detail.

### 3.4.3 Initial OH reaction site

**Scheme 1** shows that OH radical can attack different carbon sites to generate a variety of reaction products. For example, an alcohol functional group can be added to different carbon sites, producing three structural isomers of major functionalization products ($C_5H_8O_5$) (**Table 2**). The fragmentation of alkoxy radicals at the tertiary (**Scheme 1, Path A**) and secondary carbon site (**Scheme 1, Path B**) can yield two isobaric compounds of major fragmentation product ($C_4H_6O_3$) (**Table 2**). These isomeric and isobaric compounds cannot be differentiated by their accurate mass measurement and, thus, the relative importance of different reaction pathways cannot be directly inferred from the aerosol speciation data. In a prior study (*Cheng et al., 2015*), we found that the initial OH abstraction likely occurs at the tertiary carbon site of two dimethylsuccinic acids (2,2-dimethylsuccinic acid and 2,3-dimethylsuccinic acid), which are structurally similar to methylsuccinic acid with an additional methyl group. The reactivity order can be explained by the stability of the alkyl radical formed after the hydrogen abstraction by the OH radical (i.e. in the order of: tertiary > secondary > primary). Additionally, the observed reactivity order shows a higher consistency with the SAR proposed for gas-phase chemistry (*Kwok and Atkinson, 1995*) than that for the dilute solution (*Monod et al., 2008; Doussin and Monod, 2013*). This may be associated with the reaction of OH radicals at the surface of aqueous methylsuccinic acid droplets rather than in the bulk (*Cheng et al., 2015*). In this work, for the OH reaction with methylsuccinic acid, the gas-phase SAR model predicts the ratio of the hydrogen abstraction rates at the tertiary site *vs.* the secondary and primary sites to be 2.07 and 10.6, respectively (*Kwok and Atkinson, 1995*). These results suggest that the tertiary carbon site (**Scheme 1, Path A**) is preferentially attacked by the OH radical, followed by the secondary carbon site (**Scheme 1, Path B,**), and the primary carbon site (**Scheme 1, Path C**) is the least favorable.

### 3.4.4 Effect of branched methyl group on the reaction pathways

While the tertiary carbon site is more likely attacked by OH radicals (**Scheme 1, Path A**), the branched methyl group can play a role in determining the reaction mechanisms. From the aerosol speciation data, a large alcohol-to-ketone functionalization product ratio is observed. At the maximum OH exposure, which is about one oxidation lifetime, the abundance of the alcohol products is about 24 times larger than that of ketone products. One possibility for explaining this is the following: since the tertiary carbon site, which is the preferential OH reaction site, allows only the addition of an alcohol functional group, the ketone products cannot be formed via both the alkoxy and peroxy radical chemistry. Another possible explanation is that the presence of the methyl group may cause steric hindrance on arranging the two peroxy radicals into a cyclic tetroxide intermediate through the Russell and the Bennett-Summers mechanisms, which is necessary for the





formation of alcohol and ketone functionalization products (*Cheng et al., 2015*). Alternatively, the self-reaction of two peroxy radicals is preferable formation pathway for alkoxy radicals. The alkoxy radicals can then form the alcohol functionalization product via intermolecular hydrogen abstraction, which can be responsible for the secondary chemistry that possibly occurred in the aqueous particle phase. As suggested by Cheng et al. (2015), the strong hydrogen bonding among the terminal carboxyl functional groups might lower the decomposition rate of the alkoxy radical. This could make the intermolecular hydrogen abstraction by the alkoxy radicals more competitive.

This large alcohol-to-ketone functionalization product ratio has also been observed for other methyl-substituted succinic acids. In the study of OH reaction with 2,2-dimethylsuccinic acid, 2,3-dimethylsuccinic acid, at approximately one oxidation lifetime, the alcohol-to-ketone functionalization product ratio is reported to be 3.03 and 77.06, respectively (*Cheng et al., 2015*). However, a ratio close to 1 at the same oxidation lifetime has been found for succinic acid, which is the less branched counterpart (*Chan et al., 2014*). These results suggest that the presence of branched methyl group(s) in the backbone of succinic acid greatly alters the heterogeneous OH chemistry relative to succinic acid. The alkoxy radical chemistry, originating from the OH abstraction of the tertiary carbon site, appears to be an important pathway in the formation of both functionalization and fragmentation products in the OH oxidation of the methyl-substituted succinic acids.

### 3.5 Two-product oxidation kinetic model

To gain more insights into how the composition of aqueous methylsuccinic acid droplets evolve during heterogeneous OH oxidation, an oxidation kinetic model (a two-product model) coupled with an aerosol thermodynamic model (AIOMFAC) is built based on the identification of reaction products and proposed reaction mechanisms. Assuming that the identified products can well represent the overall particle composition, the proposed reaction pathways (**Scheme 1**) is used as a basis and a simplified reaction scheme (**Scheme 2**) is applied in the model (justifications are discussed below). The amount of particle-phase water together with the activity coefficients of the methylsuccinic acid and its two products are predicted using the AIOMFAC model based on the simulated distribution of the reaction products. The main objective of this model is to examine how the competition between functionalization and fragmentation processes, alongside with volatilization, determines the evolution of particle-phase reaction products and water content at different extents of oxidation. The detailed description of the model is given below.

### 3.5.1 Heterogeneous OH reaction rate

The heterogeneous reaction rate is described by the gas-aerosol collision frequency, as proposed by Cappa and Wilson (2012):

$$\frac{d[R]}{dt} = -\gamma_{\text{eff}} \, \overline{c_{OH}} \, [OH] \, \pi \, D_0^2 \, f , \tag{3}$$





where [OH], $\overline{c_{OH}}$ and $D_0$ denote the concentration of gas-phase OH radicals, mean velocity of gas-phase OH radicals, and the mean surface-weighted diameter, respectively. $f$ is the ratio between the concentration of methylsuccinic acid at reaction time ($t$) and its initial concentration (i.e. [R]/[R]$_0$). The effective uptake coefficient $\gamma_{eff}$ was obtained from the experiment via Eq. (2) and assumed to be constant during the oxidation.

**3.5.2 Reaction scheme**

Unlike explicit chemistry models described in the literature (*Wiegel et al., 2015*), a detailed reaction mechanism is not employed in this model. A few assumptions are made concerning the simplified reaction scheme (**Scheme 2**). First, only two products are considered: one major functionalization product ($C_5H_8O_5$) and one major fragmentation product ($C_4H_6O_3$). This assumption is reasonable as these two products contribute the majority of the identified products (**Fig. 2 and 3**). Second,
based on results reported in the literature and the gas-phase SAR model predictions, the OH abstraction of the hydrogen atom from the tertiary carbon site (**Scheme 1, Path A**) is more favorable than the primary (**Scheme 1, Path C**) and secondary carbon sites (**Scheme 1, Path B**) as discussed in the Sect. 3.4.3. For simplicity, the model assumes the tertiary carbon site (**Scheme 1, Path A**) as the dominant OH reaction site. It is acknowledged that isomeric and isobaric compounds likely exist for these two major products. Based on the AIOMFAC model predictions, these structural isomers and isobaric
compounds exhibit very similar hygroscopicity at 85 % RH (see **Table 2**). It is thus presumed that the choice of different combinations of isomeric and isobaric compounds does not significantly affect the model predictions of the particle-phase water and activity coefficient of the species. Third, the two major 1[st] generation reaction products are assumed to be not further oxidized by OH radicals. This is due to the observed insignificance of 2[nd] or higher generation products in the aerosol mass spectra (**Fig. 2 and 3**) under the present experimental conditions.

Overall, the simplified reaction scheme (**Scheme 2**) assumes the tertiary carbon site (i.e. **Scheme 1, Path A**) as the sole OH reaction site and could give a reasonable representation of the overall reaction pathways with sufficient details of the reaction products. The molecular yields ($\alpha_1$ and $\alpha_2$) of the two products are the only adjustable parameters of the model and are obtained by fitting the kinetic model to the aerosol speciation data (**Fig. 3**) without considering the minor products, which
contribute about 8.5 % of the total products at the maximum OH exposure.

**3.5.3 Gas-particle partitioning and gas-phase oxidation**

Oxidation can lead to the loss of particle mass through the volatilization of the fragmentation products. An absorptive gas-particle equilibrium partitioning of species $i$ is assumed to be achieved for the aqueous droplet and is expressed here by its effective saturation vapor concentration, $C_i^*$ (*Pankow, 1994; Donahue et al., 2006; Zuend and Seinfeld, 2012*):

$$C_i^* = \frac{p_{sat,i}}{RT} \frac{\gamma_i \sum_k C_k^L}{\sum_k \frac{C_k^L}{M_k}} ,$$ (4)



where $p_{sat,i}$ and $\gamma_i$ are the liquid-state pure-component saturation vapor pressure at temperature $T$, and the mole-fraction-based activity coefficient of species $i$, respectively. $R$ is the ideal gas constant, $C_k^L$ the mass concentration of mixture species $k$ in the liquid-phase of the aerosol system (units of kg m$^{-3}$ of air) and $M_k$ the molar mass, with index $k$ covering all organic species and water. Using the proposed molecular structures of the reaction products, the $p_{sat}$ of the species are estimated

using SIMPOL.1, a group contribution method developed by Pankow and Asher (2008), and the EVAPORATION model (*Compernolle et al., 2011*). For pure methylsuccinic acid, the saturation vapor pressure predicted by SIMPOL.1 and EVAPORATION at 293 K is $2.79 \times 10^{-3}$ Pa and $7.88 \times 10^{-4}$ Pa, respectively, while its experimental value is reported to be $2.54 \times 10^{-4}$ Pa (*Booth et al., 2011*). As the EVAPORATION model can reasonably well predict the vapor pressure of parent methylsuccinic acid, it is used to estimate the vapor pressure of the species in the model simulation. It is acknowledged that

different saturation vapor pressure values result in different optimized fitted yields in the model simulation. The fraction of species ($i$) remaining in the particle phase is computed by Eq. (5) (*Zuend and Seinfeld, 2012*):

$$r^{PM}_i = \left(1 + \frac{C_i^*}{\Sigma_k C_k^L}\right)^{-1}, \tag{5}$$

In addition, non-ideal interactions between the organic species and between water and organics in the particle phase influence the gas-particle partitioning. The equilibrium partial pressures of the species can be over/under-estimated by

assuming an ideal solution in the liquid phase (i.e. activity coefficients of the species, $\gamma_i$, equal to one). The extent of the over/under-prediction can vary by functional groups (e.g. alcohol, ketone, and carboxylic acid) and is affected by temperature, aerosol water content, and aerosol mass loading (*Zuend et al., 2010; Zuend and Seinfeld, 2012*). In our model, the liquid-phase non-ideality is considered explicitly by using activity coefficients of species $i$ estimated by the thermodynamic model (the computed activity coefficient of the species at different OH exposures are shown in the

Supporting Information). We acknowledge that gas-phase measurements of organic species partial pressures are not available in this study. Despite this limitation, the particle speciation data are used to provide reasonable constraints on the formation and volatilization of the fragmentation product.

The parent methylsuccinic acid and the functionalization product have low volatilities and primarily remain in the particle

phase. A fraction of the fragmentation product can partition into the gas phase. Once present in the gas phase, the fragmentation product may undergo further oxidation by OH radicals. The gas-phase reaction rate constant is estimated by the gas-phase SAR model (*Kwok and Atkinson, 1995*). For simplicity, the reaction products formed from the gas phase OH oxidation of the fragmentation product are assumed to be volatile and remain in the gas phase without partitioning back to the particle phase.

**3.5.4 Particle-phase water and activity coefficients of the species**

Given the simulated aerosol composition and the environmental conditions (e.g. RH and temperature) inside the reactor, the equilibrium aerosol water content and the activity coefficients of the species ($\gamma_i$) are computed using the AIOMFAC model,





which takes into account the molecular interactions between the species in the liquid droplets using a group contribution method.

### 3.5.5 Particle size

Upon oxidation, the size of the droplets is subject to change in response to the formation of the reaction products, the
volatilization of fragmentation products, and the condensation or evaporation of water molecules. The droplet size is allowed to vary over the entire oxidation and is calculated using an additivity rule based on known partial molar densities, molar masses and molar amounts in the liquid phase.

### 3.6 Model Results

**Figure 3** shows the comparison between the modeled and measured particle composition at different OH exposure levels.
Reasonable results can be obtained with the fitted yields: $\alpha_1 = 0.57$ and $\alpha_2 = 0.37$. The total fitted yield of the two products is 0.94 (less than one) to account for about 8 % of the other products formed at the maximum OH exposure. At low OH exposures, the model results show a relatively good agreement with the measured values. The model-experiment discrepancy increases with increasing OH exposure, which is expected as the contribution of minor products to the total products becomes more significant at higher OH exposures and further oxidation of 1st generation reaction products are not being
considered in this model. At the maximum OH exposure, the model predicts that about 65 % of the methylsuccinic acid is oxidized to form the more hygroscopic functionalization product, while about 9 % is decomposed into the fragmentation product, which is less hygroscopic than methylsuccinic acid. As shown in **Fig. 4,** model simulations can also reasonably well predict the parent compound decay.

**Figure 5** shows the simulated fractional contribution of each species to the particle volume upon oxidation. When the parent methylsuccinic acid is reacted away, the contributions of functionalization and fragmentation products to the droplet volume increase with increasing OH exposure. Water molecules contribute significantly to the total aerosol volume before and after oxidation. An increase in water volume contribution is observed along with a decrease in total volume of other non-water components (i.e. unreacted methylsuccinic acid, functionalization product, and fragmentation product). A net decrease of 5.1
25   % in the total volume fraction of all non-water components is observed at the maximum OH exposure. This is likely attributed to the fragmentation product lost via volatilization.

The evolution in particle-phase water content during oxidation was further examined. **Figure 6A** shows that when the OH exposure increases from 0 to $1.47 \times 10^{12}$ molecule cm$^{-3}$ s, the simulated mass fraction of water increases from 0.362 to 0.424.
This can be attributed partially to the finding that oxidized droplets contain a significant fraction of the functionalization product, which is more hygroscopic than the methylsuccinic acid (**Fig. 3**). On the other hand, as shown in **Fig. 6B**, the relative change in number of water molecules exhibits a different trend from that of water mass fraction (**Fig. 6A**) upon





oxidation. The number of water molecules experiences a slight increase of about 1.6 % at low OH exposures (0 to $6 \times 10^{11}$ molecule $cm^{-3}$ s) and ends with an overall decreasing trend to the loss of 7.9 % of water molecules at the maximum OH exposure. The initial increase can be explained by the higher abundance of functionalization products that enhance the particle hygroscopicity at lower OH exposures. In proportion to the amount of organic mass, the oxidized droplets thus

5   contain more water molecules than unreacted ones. As the oxidation proceeds further (i.e. to the higher OH exposures), the formation of the fragmentation product becomes more significant. The enhancement in the water content by the formation of functionalization products cannot compensate completely for the loss of water molecules via the volatilization of fragmentation products, which leads to a net decrease in the number of both particle-phase organic molecules and associated water molecules as the RH remains at a controlled value.

The simulated aerosol size change in **Fig. 7** also shows the resulting effect of the formation and volatilization of fragmentation products. The simulated diameter is predicted to decrease from 237.2 nm to 222.6 nm, which agrees reasonably well with the decreasing trend exhibited by the experimental data at low OH exposures. The largest deviation is observed at the maximum OH exposure. This could be explained by that for the particle composition (**Fig. 3**), the model-

experiment discrepancy increases with increasing OH exposure, as discussed in the preceding section. A decrease in aerosol size suggests that the aerosols showed a net loss of material at all OH exposure levels. The decrease in droplet size is mainly attributed to the volatilization of the fragmentation product and the evaporative loss of an amount of water associated with the loss of fragmentation product. It is also noteworthy that the enhancement in aerosol hygroscopicity by the formation of functionalization products at early oxidation stages can slow down the aerosol size reduction by offsetting the effect of

volatilization by the fragmentation reactions.

## 4 Conclusions

The relationship between heterogeneous oxidation and the compositional transformation of aqueous methylsuccinic acid droplets is investigated using experimental approach and model simulation. Reaction products are characterized using a soft ambient pressure ionization source (DART) coupled with a high resolution mass spectrometer. The formation of major

functionalization and fragmentation products are likely explained by the intermolecular hydrogen abstraction and unimolecular decomposition of tertiary alkoxy radicals formed at the methyl-substituted carbon site, respectively. Assuming that the composition of the aerosol is relatively well-characterized, a two-product oxidation kinetic model coupled with an aerosol thermodynamic model is developed to examine the chemical evolution of aerosol composition during oxidation. Model results show that water molecules contribute significantly to the droplet mass (and volume) before and after oxidation.

At the early oxidation stage, the net hygroscopicity of the aerosols is slightly enhanced due to the formation of more oxidized functionalization products. Although the oxidized droplets can uptake more water than unreacted ones (relative to the organic content), the number of water molecules is found to decrease at higher oxidation exposure. This is attributed to





the increased formation and volatilization of fragmentation products, which reduce the aerosol organic mass and associated amount of water. In conclusion, this study shows the relative importance of functionalization and fragmentation processes, alongside volatilization, on the evolution of the particle-phase reaction products and liquid water content, which are largely dependent on the extent of oxidation and the differences in volatilities of the organic species formed.

## 5 Acknowledgement

M. M. Chim, C.T. Cheng, and M. N. Chan are supported by the Direct Grant for Research (4053089) and One-Time Funding Allocation of Direct Grant (3132765), The Chinese University of Hong Kong. J. F. Davies are supported by the Director, Office of Energy Research, Office of Basic Energy Sciences, Chemical Sciences, Geosciences, and Biosciences Division of the U.S. Department of Energy under Contract No. DE-AC02-05CH11231. A. Zuend acknowledges support by the Natural

Sciences and Engineering Research Council of Canada (NSERC), grant RGPIN/04315-2014.

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

30

35



**Table 1. Chemical structure properties, rate constant and effective OH uptake coefficient of methylsuccinic acid.**

| | |
|---|---|
| Chemical structure | 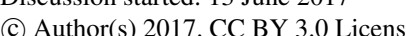 |
| Chemical Formula | $C_5H_8O_4$ |
| Oxygen-to-carbon ratio, O/C | 0.8 |
| Hydrogen-to-carbon ratio, H/C | 1.6 |
| Mean carbon oxidation state, $\overline{OS}_C$ | 0 |
| Carbon number, $N_C$ | 5 |
| Heterogeneous OH rate constant, $k$ ($\times 10^{-12}$ $cm^3$ $molecule^{-1}$ $s^{-1}$) | $1.07 \pm 0.09$ |
| Effective OH uptake coefficient, $\gamma_{eff}$ | $1.02 \pm 0.17$ |



**Table 2. Major reaction products observed during the heterogeneous OH oxidation of aqueous methylsuccinic acid droplets. The relative abundance is reported at the maximum OH exposure ($1.47 \times 10^{12}$ molecule cm$^{-3}$ s).**

| Chemical Formula | Molecular Weight | Relative Abundance (%) | Proposed Chemical Structure |
|---|---|---|---|
| $C_5H_8O_4$ *Parent methylsuccinic acid* | 131 | 30.3 | **Water Mole Fraction[#]** = 0.806<br>$p_{sat,\ SIMPOL}$[*] = $2.79 \times 10^{-3}$ Pa<br>$p_{sat,\ EVAPORATION}$[*] = $7.88 \times 10^{-4}$ Pa |
| $C_5H_8O_5$ *Functionalization Product* | 147 | 48.4 | Tertiary / Secondary / Primary<br>**Water Mole Fraction[#]** = 0.853 (Tertiary)<br>= 0.857 (Secondary)<br>= 0.865 (Primary)<br>$p_{sat,\ SIMPOL}$[*] = $1.63 \times 10^{-5}$ Pa<br>$p_{sat,\ EVAPORATION}$[*] = $2.24 \times 10^{-5}$ Pa |
| $C_4H_6O_3$ *Fragmentation Product* | 101 | 12.8 | Tertiary / Secondary<br>**Water Mole Fraction[#]** = 0.781 (Tertiary)<br>= 0.771 (Secondary)<br>$p_{sat,\ SIMPOL}$[*] = 3.38 Pa (Tertiary)<br>= 1.30 Pa (Secondary)<br>$p_{sat,\ EVAPORATION}$[*] = 1.87 Pa (Tertiary)<br>= 2.96 Pa (Secondary) |

[*] $p_{sat}$ = Saturation vapor pressure estimated by the SIMPOL.1 model and EVAPORATION model.

[#] The corresponding water mole fractions in aqueous solutions of the different (pure) structural isomers at vapor-liquid equilibrium with 85 % RH (i.e. at 85 % water activity) as predicted by the AIOMFAC model.





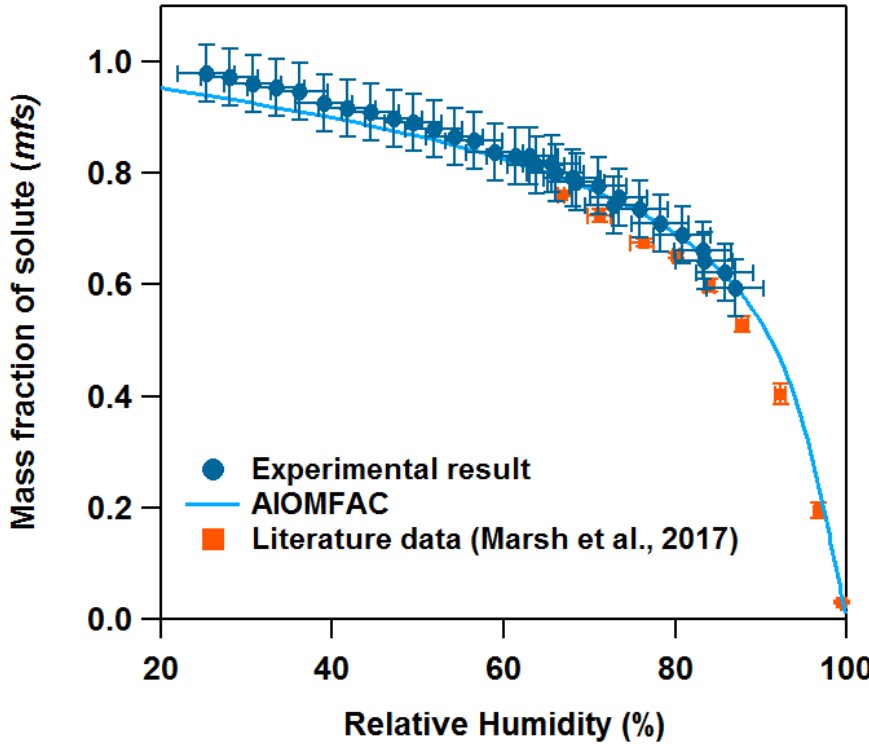

**Figure 1. Hygroscopicity of methylsuccinic acid droplets (~ 10 µm diameter) measured using an aerosol optical tweezers setup at a temperature of 295 K and corresponding prediction by the AIOMFAC model. The orange markers denote experimental hygroscopicity data by Marsh et al. (2017).**





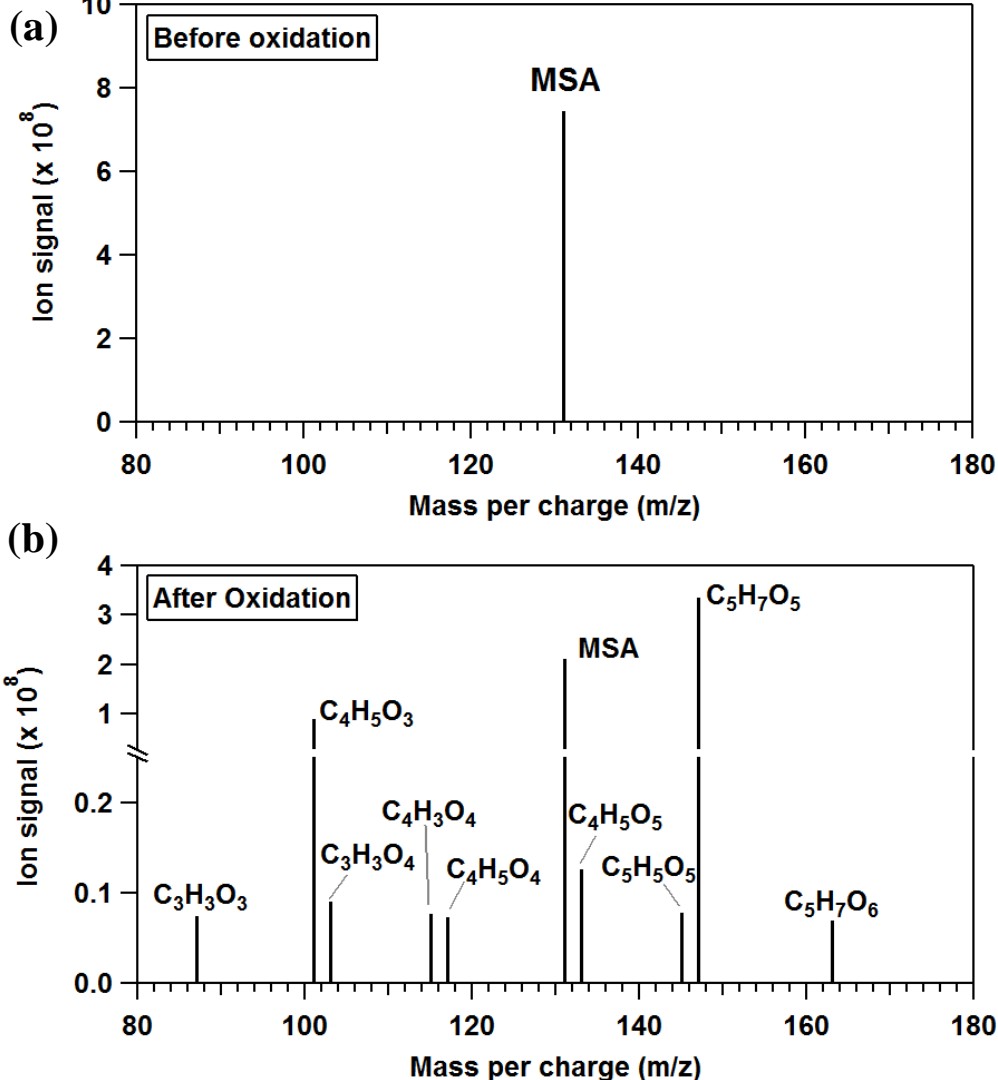

**Figure 2.** Aerosol mass spectra (a) before, and (b) after heterogeneous OH oxidation of aqueous methylsuccinic acid (MSA) droplet. The minor products after oxidation contribute less than 10 % of the total ion signal.





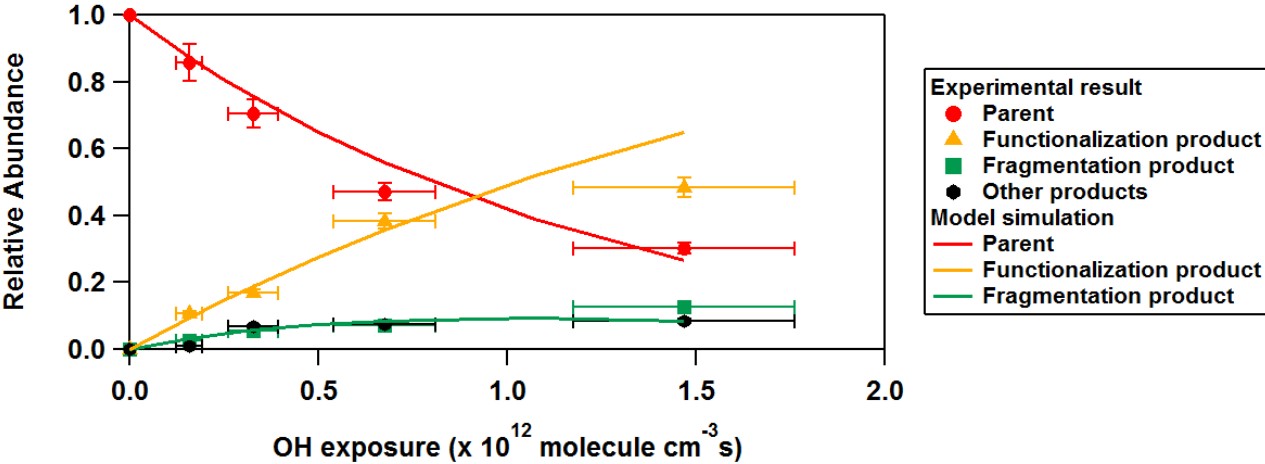

**Figure 3. Evolution of the parent ($C_5H_8O_4$), major functionalization ($C_5H_8O_5$) and major fragmentation ($C_4H_6O_3$) products during heterogeneous OH oxidation of aqueous methylsuccinic acid droplet. Experimental values are shown in markers with one standard deviation as uncertainty.**

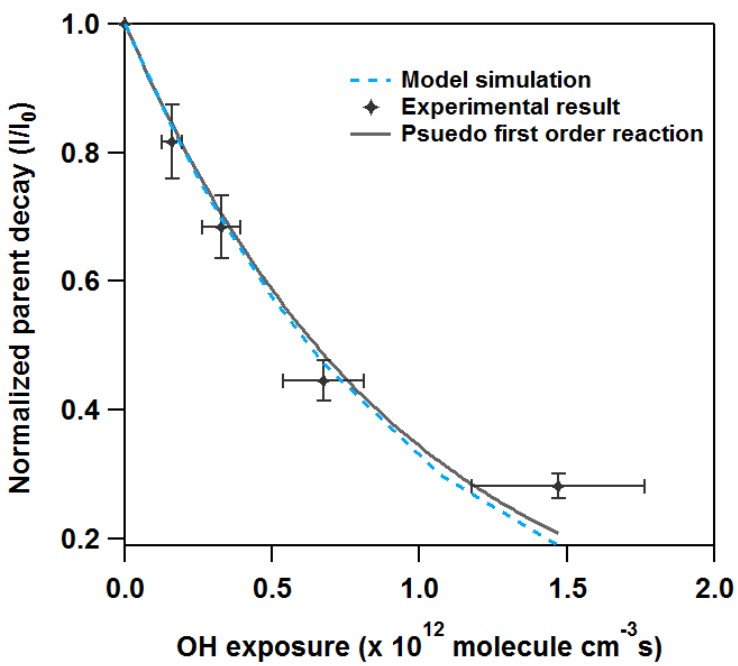

**Figure 4. Normalized parent compound decay from experimental results and model simulations.**



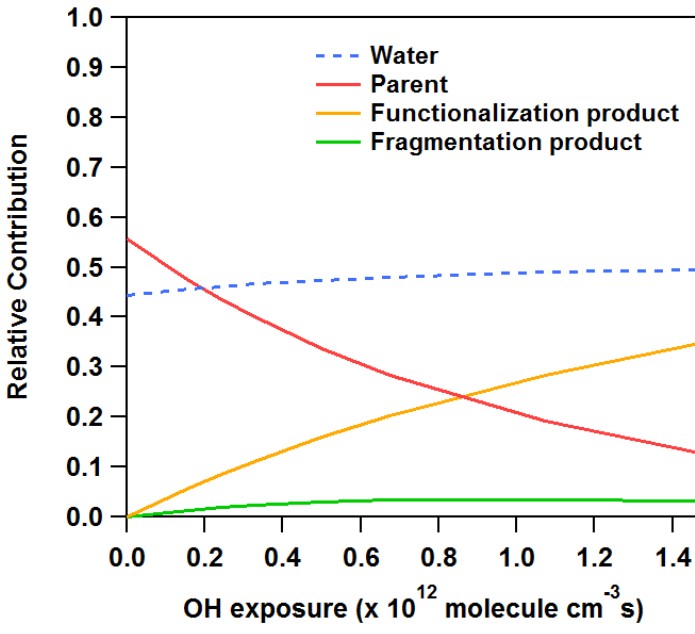

**Figure 5. Simulated relative contribution of each species to the particle volume during heterogeneous OH oxidation of aqueous methylsuccinic acid droplet at 85 % RH.**

5 **(a)** **(b)**

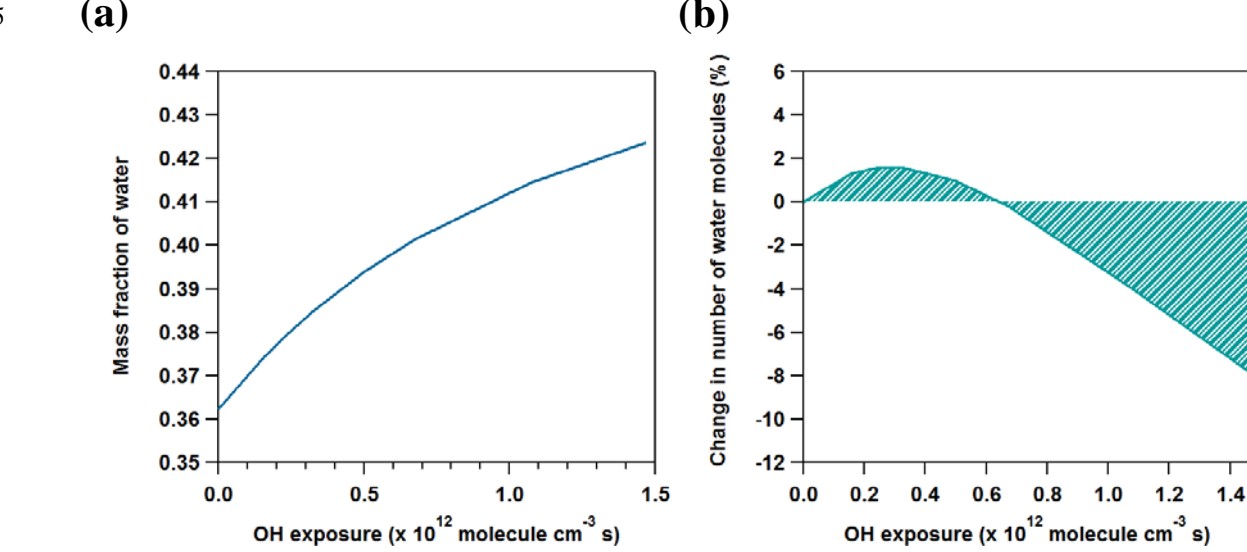

**Figure 6. Simulated evolution of aerosol water content in terms of (a) water mass fraction, and (b) the percentage change in number of water molecules, during heterogeneous OH oxidation of aqueous methylsuccinic acid droplet.**





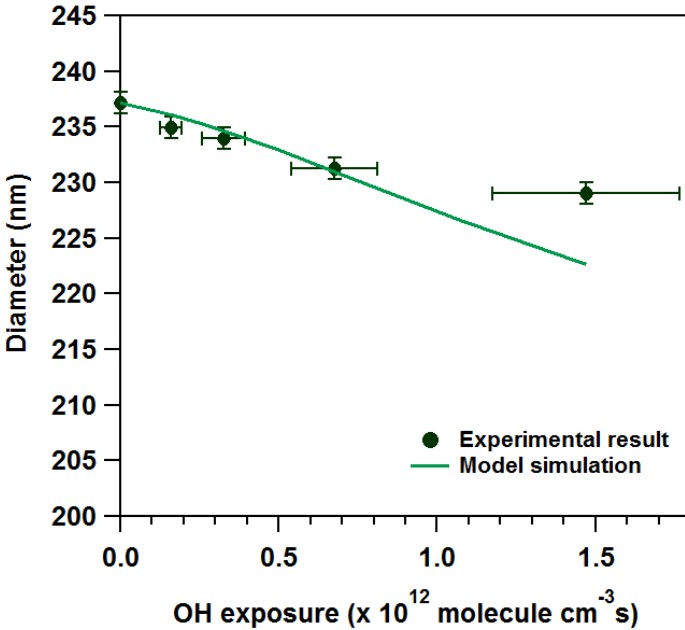

**Figure 7. Simulated and experimental result of aerosol diameter during heterogeneous OH oxidation of aqueous methylsuccinic acid droplet.**





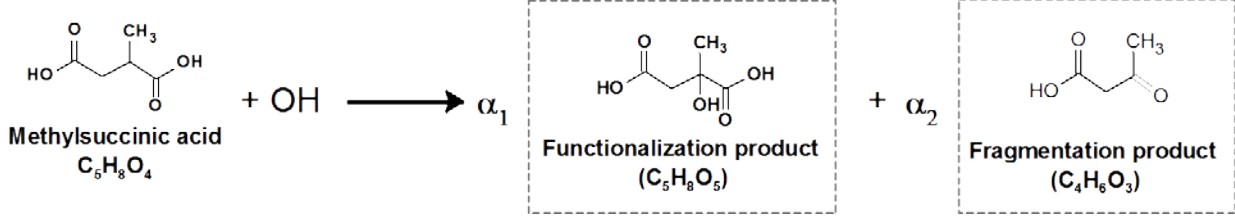

**Scheme 1. Proposed reaction scheme for the 1$^{st}$ generation products of the heterogeneous OH oxidation of aqueous methylsuccinic acid droplet.**

**Scheme 2. Simplified reaction scheme used in the model simulation (with $\alpha_1 = 0.57$ and $\alpha_2 = 0.37$ determined from optimized model-measurement comparison).**