# Peer review of "Compositional Evolution of Particle-Phase Reaction Products and Water in the Heterogeneous OH Oxidation of Model Aqueous Organic Aerosols"

_Atmospheric Chemistry and Physics, 2017_

## Referee Comment (RC1) · Anonymous Referee #3 · 4 Jul 2017

This is an interesting paper that has lots of interesting aspects to it. Two different experimental approaches are used: laser tweezers for aerosol hygroscopicity measurements and DART MS for aerosol oxidation. The experimental results are understood using extensive chemical mechanism prediction and a kinetic model which uses thermodynamic input from the AIOMFAC model. However, the paper has a number of deficiencies that need to be tackled before it can be published. There are lots of details missing that are required for subsequent researchers to replicate the experiments. And the methodology makes some big assumptions that need to further explored.

Major Comments Since this paper is in an atmospheric journal, it needs to acknowl-

edge that the oxidation experiments were run under non-atmospheric conditions. In particular, the OH concentrations were very far from being atmospherically relevant, even if the exposures are reasonable. What possible consequences are there of using the very high OH concentrations?

What is the size distribution of the aerosols in the flow tube experiments? An average size is stated (P7 L 19, 237.2 nm diameter) but no details of the size of the distribution is given. This is important since size has been shown to be important for aerosol reactivity as shown in Al Kindi et al. (2016). The work of Marcoli et al. (2004) suggests that the deliquescence point of methyl succinic acid is 95%, which is significantly higher than the RH used in the heterogeneous oxidation experiments. How did you ensure the particles were aqueous? Do you disagree with the results of Marcoli et al.?

The ionization efficiency of all oxidation products is assumed to be the same as methyl succinic acid (P7 line 3). This is a big assumption. Previous MS work has shown that ionization efficiencies vary massively. For example, in Al Kindi et al. (2016) there is approximately a factor of 20 between difference dicarboxylic acids and mono-carboxylic acids, albeit whilst using a different type of MS. The Orbitrap approach has also shown big differences in ionization efficiency as well – see the work and references of Markus Kalberer in Cambridge including Gallimore et al. (2011). Since the simplified reaction scheme uses products with 3 and 1 carboxylic acid groups, I really worry how the experimental results with the huge uncertainties in ionization efficiency can be fed into this model. This point has potential repercussions throughout the paper especially with respect to the modelling and reaction mechanism sections. For example, on P8 L26 "The low abundance of smaller products is presumably due to their higher volatility..." or it could be that their ionization efficiency is much smaller because of a lack of carboxylic acid and other functional groups. Another example is on P13 L10, the fitted yields will be very different if the ionization efficiencies are significantly different to each other. These repercussions need to be made very clear and more justification of the approach is required. I understand that standards might not be available but if

the relative concentrations cannot be known then it makes for very weak foundations for the rest of the paper.

In the conclusion it needs to be emphasized that a highly idealized system was investigated. This system can potentially provide insights into atmospheric chemistry but it does not mimic real atmospheric chemistry due to the atmospherically unrealistic single component system, high OH concentrations, etc.

Minor comments Title – should acknowledge that the paper investigates the OH oxidation of methyl succinic acid not organic species in general. Suggested title "Composition evolution of particle phase reaction products and water in the heterogeneous OH oxidation of aqueous methyl succinic acid droplets"

Abstract – line 16 change "at/near" to "at or near"

Intro – somewhere in the intro it should be noted that methyl succinic acid is a deliquescent compound, see Marcolli et al. (2004)

Intro – line 4 "This radical initiated heterogeneous oxidation. . ." Previous sentence talks about ozone as well as OH. Ozone is not a radical species.

Intro P2 – line 5 ". . .change the aerosol composition and, therefore, alters the properties of aerosols.." This statement is too strong. I think the jury is still out about how important organic oxidation is for general aerosol properties.

Intro P2 – lines 15-20 – there is a nice description of the roles of water in organic aerosol chemistry in Gallimore et al. (2011). The water can modify the viscosity/diffusion. It can also act as a reactant, this is in comparison to the reference given in the text Chim et al. (2017). The Gallimore reference is for ozonolysis of organic species but the water could still potentially interact with the radical chemistry.

Intro P3 – line 2 – changing composition can also change the deliquescence and efflorescence points of aerosol.

P4 – lines 4-5 – need to make clear that methyl succinic acid whilst one of the most abundant components in aerosol, its concentration is still low compared to all the other species. i.e. understanding the reactivity of one species does not give you a good understanding of the total aerosol composition reactivity.

P4 Line 17 what concentration of ozone was used? Note at high concentrations, ozone can react with moieties other than double bonds.

P4 L27 – how do you know that all particles are vaporized at 250 ïĆřC? What is the evidence? Or if appropriate, provide a reference.

P5 Line 3 – this seems like an excessive number of references for one technique. What do the different papers add? Which one is the key reference?

P5 Line 5 – How good was the mass calibration, how accurately can you define the different reaction products?

P5 hygroscopicity measurements – how many particles were measured? It sounds like only one particle was measured? If this is the case, then that is a little cavalier but the agreement with Marsh et al. gives confidence. What do the error bars in Figure 1 represent?

P6 L4-4 the assumption that AIOMFAC is good predictor of oxidation produce hygroscopicity because it correctly predicts the hygroscopicity of methyl succinic acid seems quite big. It would be good to reference papers that look at the general performance of AIOMFAC. Does AIOMFAC make good predictions of species with similar functional groups as the oxidation products measured in this paper?

P6 L15-23 a long justification is given for possible differences between the CK-EDB and laser tweezer experiment. Since the two measurements agree with each other, within experimental errors, I'm not sure the justification is required?

P7 Line 22. The error bars on the gamma value need to be acknowledged. I think the large error bars preclude a discussion of the gamma likely being greater than 1. For

OH, a gamma of 1 is expected.

P11 Line 3 When the error in the gamma measurement is propagated through the model what is the outcome?

References

Marcolli, Claudia, Beiping Luo, and Thomas Peter. "Mixing of the organic aerosol fractions: Liquids as the thermodynamically stable phases." The Journal of Physical Chemistry A 108.12 (2004): 2216-2224.

Al-Kindi, S.S., Pope, F.D., Beddows, D.C., Bloss, W.J. and Harrison, R.M., 2016. Size-dependent chemical ageing of oleic acid aerosol under dry and humidified conditions. Atmospheric Chemistry and Physics, 16(24), pp.15561-15579.

Gallimore, P.J., Achakulwisut, P., Pope, F.D., Davies, J.F., Spring, D.R. and Kalberer, M., 2011. Importance of relative humidity in the oxidative ageing of organic aerosols: case study of the ozonolysis of maleic acid aerosol. Atmospheric Chemistry and Physics, 11(23), pp.12181-12195.

---

## Referee Comment (RC2) · Anonymous Referee #2 · 7 Jul 2017

The authors present experimental results on the OH oxidation of methyl succinic acid particles, which were generated from solution and maintained as aqueous particles at high humidity (RH=85%). Accompanying measurements of the water content of methyl succinic acid particles under varying humidity conditions are also presented. Experimental results for the oxidation rate of the methyl succinic acid and the hygroscopicity of methyl succinic acid are then used to model the changes in composition and size of methyl succinic acid particles as they are oxidized by OH. The model uses simplified chemistry (only two oxidation products and no secondary oxidation) with a sophisticated thermodynamic model (AIOMFAC) that predicts water content/particle size taking into account the changing activity coefficients of the methyl succinic acid,

the two oxidation products, and water activity. The major conclusions are that oxidized methyl succinic acid particles increase in hygroscopicity and lose a notable amount of organic mass due to formation of volatile fragmentation products. The work presented is clearly described and presented fairly well. Understanding the oxidation of water soluble organic material, such as the di-acid presented here, is important in understanding the effects of water on particle oxidation processes. I recommend publication pending minor revisions.

General Comments

The authors must clarify the range of oxidation conditions. The high levels of oxidation are equivalent to a week in the atmosphere even for a moderate to high level of OH (2E6 mol/cmˆ3). A time axis for a given atmospheric OH level should be added to Figure 3.

A better sense of the mass balance during the experiments is needed, particularly for the volatilization. Some analysis of the amount of organic material lost to the gas phase must be presented. Using a simple mass balance and the assumed (or known) product hygroscopicities, how much fragmentation/volatilization is suggested given your experimental results? The simplified chemical mechanism is hard to reconcile with the results in Figure 3, because it would appear that the ratio of the products should always be (0.57/0.37). It seems like the predicted decrease in particle size would require a greater extent of fragmentation/volatilization. In any case the results in Figure 3 should be more clearly discussed in terms of the simplified chemical mechanism.

The linearity of your mass spectrometry measurements should be shown. All the analysis presented assumes that the response of your mass spectrometer is linear over the wide range of MSA and product concentrations. A calibration using known amounts of MSA, and the linearity of response up to signal levels of 10ˆ9, should be shown.

Specific Comments Page 7 Line (10) It should be directly stated that this rate constant is an effective rate constant for OH radicals with aqueous methyl succinic acid particles.

It needs to be clear that the rate constant for OH + MSA was not directly measured in a single phase. 14 (5) "As the oxidation proceeds further (i.e. to the higher OH exposures), the formation of the fragmentation product becomes more significant " It is not clear how the model accounts for this. Do the alpha values change during the course of the oxidation? Is secondary oxidation of the functionalization and fragmentation products taken into account? It seemed that the model description specifically does not include the secondary oxidation.

14(12) " The largest deviation is observed at the maximum OH exposure. This could be explained by that for the particle composition (Fig. 3), the model-experiment discrepancy increases with increasing OH exposure, as discussed in the preceding section." This point is nearly lost in the awkward sentence construction. Please re-word such as: "The large deviation in particle size observed at the maximum OH exposure can be explained by the poorly predicted particle composition at high OH exposure (Fig. 3)." 14(30) "net hygroscopicity of the aerosols is slightly enhanced due to the formation of more oxidized functionalization products." Hygroscopicity is an intensive property, so the term "net" hygroscopicity lacks a clear meaning. In fact you present that the hygroscopicity of the particle organic content increases. In other words, the activity of water is further suppressed. The particles lose water due to loss of mass of soluble material. Please remove the term "net hygroscopicity" and re-word this sentence more similarly to Page 1 line 33.

---

## Author Comment (AC1) · 20 Sep 2017

*This is an interesting paper that has lots of interesting aspects to it. Two different experimental approaches are used: laser tweezers for aerosol hygroscopicity measurements and DART MS for aerosol oxidation. The experimental results are understood using extensive chemical mechanism prediction and a kinetic model which uses thermodynamic input from the AIOMFAC model. However, the paper has a number of deficiencies that need to be tackled before it can be published. There are lots of details missing that are required for subsequent researchers to replicate the experiments. And the methodology makes some big assumptions that need to further explored.*

We would like to sincerely thank the reviewer for his/her thoughtful comments and suggestions. Please see our responses to reviewer's comments and suggestions below.

**Major Comments**

*"Since this paper is in an atmospheric journal, it needs to acknowledge that the oxidation experiments were run under non-atmospheric conditions. In particular, the OH concentrations were very far from being atmospherically relevant, even if the exposures are reasonable. What possible consequences are there of using the very high OH concentrations?"*

**Author Response:**

We agree with the reviewer's comments and have discussed the possible consequences of using the very high OH concentrations. The OH concentrations may have implications on the chemistry. At high OH concentrations, a significant amount of peroxy ($RO_2$) radicals are likely generated owing to the rapid oxidation of organic compounds. As such, the bimolecular $RO_2$ + $RO_2$ reactions can be favorable. However, under atmospheric OH conditions, other reactions such as $HO_2$ + $RO_2$, $NO$ + $RO_2$ and $NO_2$ + $RO_2$ may be important at lower $RO_2$ concentrations (*Wiegel et al., 2015, 2017*). To date, the effects of OH concentrations on the heterogeneous oxidation kinetics and chemistry have not fully investigated and remain largely uncertain. Additional experimental and model studies are warranted to investigate these effects. This information is added in section 2.1 Heterogeneous Oxidation of the manuscript.

Page 5 Line 20, "The OH concentrations may have implications on the heterogeneous chemistry. At high OH concentrations, a significant amount of peroxy ($RO_2$) radicals are likely generated owing to the rapid oxidation of organic compounds, which favours the bimolecular $RO_2$ + $RO_2$ reaction. Under atmospheric OH condition, other reactions such as $HO_2$ + $RO_2$, $NO$ + $RO_2$ and $NO_2$ + $RO_2$ may be important at the lower $RO_2$ concentrations. To date, the effects of OH

concentrations on the heterogeneous oxidation kinetics and chemistry have not been fully investigated and remain largely uncertain. Additional experimental and modelling studies are warranted to investigate these effects."

*"What is the size distribution of the aerosols in the flow tube experiments? An average size is stated (P7 L 19, 237.2 nm diameter) but no details of the size of the distribution is given. This is important since size has been shown to be important for aerosol reactivity as shown in Al Kindi et al. (2016). The work of Marcoli et al. (2004) suggests that the deliquescence point of methyl succinic acid is 95%, which is significantly higher than the RH used in the heterogeneous oxidation experiments. How did you ensure the particles were aqueous? Do you disagree with the results of Marcoli et al.?"*

**Author Response:**

We have added the geometric standard deviation of particle size distribution in the text. As suggested by the reviewer, we also acknowledge that the particle size can play a role in determining the aerosol reactivity, which in turn governs the composition of the aerosols.

Page 9 Line 7, "Prior to oxidation, the mean surface-weighted droplet diameter was determined to be $D_0 \sim 237.2$ nm with a geometric standard deviation of 1.52 nm. It is worthwhile to note that particle size can play a role in governing the aerosol reactivity. For instance, Al Kindi et al. (2016) showed that the distribution of reaction products greatly depends on the particle size for the heterogeneous ozone reaction with oleic acid."

We would like to thank the reviewer for bringing up the paper by Marcolli et al. (2004) concerning the water-solubility of methylsuccinic acid. The work by Marcolli et al. (2004) suggests that when the methylsuccinic acid aerosols are initially solid particles, they will deliquesce at 95.5 % RH upon humidification (at a temperature of ~298 K). From our hygroscopicity measurements for dehumidification conditions, we conclude that the aqueous methylsuccinic acid droplets do not crystallize while dehydrating to RH as low as 20 %. In our oxidation experiments, aqueous methylsuccinic acid droplets generated from the atomizer were directly mixed with humidified nitrogen, oxygen, and ozone before introduction into the aerosol flow tube reactor (at high RH). After mixing, RH and temperature inside the aerosol flow tube reactor were controlled to remain at 85 % and 20 $^{\circ}$C, respectively. Since the RH inside the reaction system is always kept substantially higher than 20 %, the methylsuccinic acid aerosols very likely existed as aqueous droplets prior to oxidation.

Page 6 Line 12, "Methylsuccinic acid is a compound with a deliquescence point at 95.5 % RH upon humidification if the aerosols are initially solid (*Marcolli et al., 2004*). In the hygroscopicity measurements, methylsuccinic acid droplets absorb or desorb water in a reversible manner in response to the set environmental RH. The aqueous methylsuccinic acid droplets maintained a spherical shape over the entire experimental RH range, and did not crystallize while dehydrating to RH as low as 20 %. As discussed in Sect. 2.1, during the aerosol flow-tube reactor experiments, aqueous methylsuccinic acid aerosols were always exposed to a sufficiently high RH of 85 % and, thus, very likely remained in a liquid state prior to oxidation."

*"The ionization efficiency of all oxidation products is assumed to be the same as methylsuccinic acid (P7 line 3). This is a big assumption. Previous MS work has shown that ionization efficiencies vary massively. For example, in Al Kindi et al. (2016) there is approximately a factor of 20 between difference dicarboxylic acids and mono-carboxylic acids, albeit whilst using a different type of MS. The Orbitrap approach has also shown big differences in ionization efficiency as well – see the work and references of Markus Kalberer in Cambridge including Gallimore et al. (2011). Since the simplified reaction scheme uses products with 3 and 1 carboxylic acid groups, I really worry how the experimental results with the huge uncertainties in ionization efficiency can be fed into this model. This point has potential repercussions throughout the paper especially with respect to the modelling and reaction mechanism sections. For example, on P8 L26 "The low abundance of smaller products is presumably due to their higher volatility: : :" or it could be that their ionization efficiency is much smaller because of a lack of carboxylic acid and other functional groups. Another example is on P13 L10, the fitted yields will be very different if the ionization efficiencies are significantly different to each other. These repercussions need to be made very clear and more justification of the approach is required. I understand that standards might not be available but if the relative concentrations cannot be known then it makes for very weak foundations for the rest of the paper."*

**Author Response:**

We agree with the reviewer's concern about the ionization efficiencies of the species measured by the DART MS. We have not measured and compared the ionization efficiencies of methylsuccinic acid and its reaction products. In an attempt to better understand the ionization efficiencies of the species, in previous study (*Chan et al., 2014*), we have quantified the composition of succinic acid droplets before and after OH oxidation from the analysis of the DART mass spectra using commercially available standards (i.e. oxalic acid, malonic acid, malic acid, oxosuccinic acid, and tartaric acid). The ionization efficiencies of oxalic acid, malonic acid, oxosuccinic acid, and tartaric acid relative to succinic acid are measured to be 0.5, 0.55, 4.59, and 2.66, respectively. The results from this study showed that the ionization efficiency can vary from 0.5 to 4.59 due to the change in carbon number ($C_2$ to $C_4$) and the addition of polar functional groups (alcohol and ketone).

As the structure and functionalities of these standards are similar to those of the reaction products proposed in this work, to better quantify the aerosol composition, we employ the ratios developed from the study of succinic acid. We assume that the ionization efficiency of methylsuccinic acid would not differ from succinic acid significantly due to their similar chemical structure. A correction factor of 1 is thus applied for methylsuccinic acid. Tartaric acid (with hydroxyl group additions) and oxosuccinic acid (with ketone functional group) are used to correct the ionization efficiencies of the $C_4$ and $C_5$ alcohol and ketone products, respectively. The malonic acid ($C_3$ dicarboxylic acid) and tartaric acid ($C_4$ dicarboxylic acid) standards are used to correct the ionization efficiency of $C_3$ and $C_4$ fragmentation products. However, we acknowledge that the ionization efficiencies of some $C_3$ and $C_4$ fragmentation products, which only contain one carboxylic acid group, have not been experimentally determined and cannot be well constrained by the standards applied here.

After correcting for the ionization efficiencies, we find that the relative abundance of the species have changed (as shown in the table below). The relative abundance before and after correcting for ionization efficiencies against OH exposure are plotted in **Fig. 3** for comparison. Generally, the abundance of the functionalization products decreases while that of parent methylsuccinic acid increases.

The use of ionization efficiencies of the standards may serve as an upper estimate of the relative abundance given their similar structure and functionalities to methylsuccinic acid and its reaction products. However, the ionization efficiencies used for correction may not correctly represent the real ionization efficiencies of methylsuccinic acid and its reaction products. We therefore decide to keep the correction factor for relative ionization efficiency as 1 for the parent methylsuccinic acid and the reaction products in this study. The relative abundance before and after correcting for ionization efficiencies can be viewed as the upper and lower estimates of the relative abundances. The alpha values are re-fitted in order to reproduce the evolution of aerosol composition during heterogeneous oxidation. We have added this information to the revised manuscript and have revised the manuscript accordingly and the changes are highlighted below.

| Chemical Formula | Relative abundance after correcting for ionization efficiencies (%) |
| --- | --- |
| $C_5H_8O_4$
*Parent methylsuccinic acid* | 50.9 |
| $C_5H_8O_5$
*Functionalization product* | 30.5 |
| $C_4H_6O_3$
*Fragmentation product* | 8.1 |

[revised manuscript text omitted]

*"In the conclusion it needs to be emphasized that a highly idealized system was investigated. This system can potentially provide insights into atmospheric chemistry but it does not mimic real atmospheric chemistry due to the atmospherically unrealistic single component system, high OH concentrations, etc."*

**Author Response:**

We add this information in the revised manuscript.

Page 16 Line 24, "The simple model system investigated in this work provides a molecular-level insight into atmospheric heterogeneous chemistry and effects on hygroscopicity. However, care must be taken in extrapolating to atmospheric conditions due to the greater compositional complexity and much lower OH concentrations of the atmosphere."

**Minor Comments**

*Title – should acknowledge that the paper investigates the OH oxidation of methyl succinic acid not organic species in general. Suggested title "Composition evolution of particle phase reaction products and water in the heterogeneous OH oxidation of aqueous methyl succinic acid droplets"*

**Author Response:**
The motivation of this paper is not to understand specifically methylsuccinic acid but to use it as a model system for aqueous atmospheric aerosols. We have changed the title of the paper to "Compositional Evolution of Particle-Phase Reaction Products and Water in the Heterogeneous OH Oxidation of Model Aqueous Organic Aerosols".

*Abstract – line 16 change "at/near" to "at or near"*

**Author Response:**
We have made the change in the abstract.

*Intro – somewhere in the intro it should be noted that methyl succinic acid is a deliquescent compound, see Marcolli et al. (2004)*

**Author Response:**
We would like to note that some single component organic particles (e.g. malonic acid, citric acid, and tartaric acid) do not crystallize when dried from high RH to low RH; even at low RH of ~10 %. From our hygroscopicity measurements under dehydration conditions, the methylsuccinic acid droplets do not crystalize down to low relative humidity (< 20 %). Our result suggests that methylsuccinic acid is likely a non-efflorescent compound in the relevant RH range of our study. We have added this information in the revised manuscript. Please refer to the response to Referee 2 for the changes.

*Intro – line 4 "This radical initiated heterogeneous oxidation: : :" Previous sentence talks about ozone as well as OH. Ozone is not a radical species.*

**Author Response:**
We have revised the sentence and removed the phase "radical initiated".

Page 2 Line 4, "This heterogeneous oxidation of organic aerosols is an important aging process …"

*Intro P2 – line 5 "can significantly change the aerosol composition and, therefore, alters the properties of aerosols.." This statement is too strong. I think the jury is still out about how important organic oxidation is for general aerosol properties.*

**Author Response:**
We have revised the sentence in the manuscript.

Page 2 Line 4, "This heterogeneous oxidation of organic aerosols is an important aging process that can change the aerosol composition and, therefore, may alter the properties of aerosols, such as their light scattering ability, hygroscopicity, and cloud condensation nuclei (CCN) activity."

*Intro P2 – lines 15-20 – there is a nice description of the roles of water in organic aerosol chemistry in Gallimore et al. (2011). The water can modify the viscosity/ diffusion. It can also act as a reactant, this is in comparison to the reference given in the text Chim et al. (2017). The Gallimore reference is for ozonolysis of organic species but the water could still potentially interact with the radical chemistry.*

**Author Response:**
We have added this information in the revised manuscript.

Page 2 Line 21, "Water can also act as a reactant and could potentially interact with the heterogeneous chemistry. For instance, Gallimore et al. (2011) observed that the distribution of the reaction products is largely dependent on the RH (affecting the aerosol water content) for the heterogeneous ozone reaction with maleic acid."

*Intro P3 – line 2 – changing composition can also change the deliquescence and efflorescence points of aerosol*

**Author Response:**
We have revised the manuscript.

Page 3 Line 3, "The formation of oxygenated functionalization products of increased water solubility can enhance the hygroscopicity property of the aerosols at a certain RH. Furthermore, the change in the composition can also change the RH and water content at which the particles undergo deliquescence or efflorescence phase transitions."

*P4 – lines 4-5 – need to make clear that methylsuccinic acid whilst one of the most abundant components in aerosol, its concentration is still low compared to all the other species. i.e. understanding the reactivity of one species does not give you a good understanding of the total aerosol composition reactivity.*

**Author Response:**
We have revised the sentences in the manuscript.

Page 4 Line 8, "Methylsuccinic acid is one of the most abundant branched dicarboxylic acids observed in atmospheric aerosols (*Li et al., 2015; Kundu et al., 2016*) and is chosen as a model compound to gain a more fundamental understanding of the heterogeneous OH chemistry of methyl-substituted dicarboxylic acids (**Table 1**). It is also worthwhile to note that the heterogeneous reactivity of atmospheric organic aerosols could differ from those observed in simple model systems owing to the complexity of the ambient aerosol composition with a typically much broader representation of organic molecules and classes of functional groups."

*P4 Line 17 what concentration of ozone was used? Note at high concentrations, ozone can react with moieties other than double bonds.*

**Author Response:**
The maximum ozone concentration used in this study was about 6.5 ppm. In separate experiments, no compositional changes are observed in the presence of ozone and the absence of UV light, suggesting that the reaction of methylsuccinic acid with ozone is not significant. We have also found that particles did not change in their composition when the UV lights were turned on in the absence of ozone. This suggests that methylsuccinic acid is not likely photolyzed. We have added this information in the Heterogeneous Oxidation section in the manuscript.

Page 4 Line 30, "The maximum ozone concentration used in this study was about 6.5 ppm. Control experiments have been done in the presence of ozone without the UV light, and in the absence of ozone with the UV light. Under both experimental conditions, no compositional

changes are observed for the methylsuccinic acid droplets, suggesting that the reaction of methylsuccinic acid with ozone is not significant and that methylsuccinic acid is not likely photolyzed."

*P4 L27 – how do you know that all particles are vaporized at 250 $^{o}$C? What is the evidence? Or if appropriate, provide a reference.*

**Author Response:**
We have not experimentally measured the size of the particles leaving the aerosol heater. We have developed an aerosol evaporation model, which has been used to predict the evaporation of different types of dicarboxylic acids particles in the ionization region (*Chan et al., 2013*). With the knowledge of the vapor pressure of the methylsuccinic acid and experimental conditions applied in the aerosol heater, the methylsuccinic acid aerosols are predicted to be fully vaporized inside the aerosol heater.

*P5 Line 3 – this seems like an excessive number of references for one technique. What do the different papers add? Which one is the key reference?*

**Author Response:**
These references are listed to show that a variety of organic species (e.g. alcohols and monoacids (*Nah et al., 2013*), dicarboxylic acid (*Chan et al., 2013, 2014*), multifunctional acids (*Cheng et al., 2016*) and methyl-substituted dicarboxylic acid (*Cheng et al., 2015, Chim et al., 2017*) can be efficiently detected by the DART. In the revised the manuscript, we only list the key references of Cody et al. (2015) and Cody (2008), which describes the working principle of the DART.

*P5 Line 5 – How good was the mass calibration, how accurately can you define the different reaction products?*

**Author Response:**
In this study, a mass tolerance is set to less than ±5 mDa for assigning the chemical formula of the detected ions. This information is added in the revised manuscript.

Page 5 Line 13, "Mass spectra were collected at 1 s intervals over a scan range from mass-to-charge ($m/z$) ratios 70–500, with each spectrum averaged over a 5-min sampling time with a mass resolution of 140,000 and a mass tolerance less than ± 5 mDa is used to assign the chemical formula of the detected ions."

*P5 hygroscopicity measurements – how many particles were measured? It sounds like only one particle was measured? If this is the case, then that is a little cavalier but the agreement with Marsh et al. gives confidence. What do the error bars in Figure 1 represent?*

**Author Response:**
The hygroscopicity measurements were performed across several droplets. Inter-droplet variability is not expected and was not observed due to the identical composition and similar size range. The factors that are most important to consider are hysteresis in the hygroscopicity, which was assessed from RH cycling up and down (and not observed for these samples in the range

from > 85 % RH to ~20 % RH, starting with wet, high-RH droplets), and the accuracy and reproducibility of the RH measurements, which are accounted for by multiple RH cycles (2 − 3 usually).

The uncertainties associated with these measurements are shown in **Fig. 1**. The x–error bars represent the uncertainty in the RH from the RH probe, and the y–error comes from the error in fitting a linear slope to the composition–refractive index calibration data. Other uncertainties in the experiment, such as in the droplet refractive index, are minor by comparison.

*P6 L4-4 the assumption that AIOMFAC is good predictor of oxidation produce hygroscopicity because it correctly predicts the hygroscopicity of methyl succinic acid seems quite big. It would be good to reference papers that look at the general performance of AIOMFAC. Does AIOMFAC make good predictions of species with similar functional groups as the oxidation products measured in this paper?*

**Author Response:**
A good model prediction for the hygroscopicity of methylsuccinic acid aerosols does not necessarily imply the same level of accuracy for oxidation products of methylsuccinic acid and their aqueous mixtures. Our confidence in AIOMFAC as a reliable model for the prediction of the hygroscopicity of methylsuccinic acid aerosols *after* oxidation relies on the following two related aspects: (1) the model's group contribution approach, which provides predictability for compounds of similar chemical structure to those used in the estimation of the adjustable model parameters, and (2) the fact that the AIOMFAC model in its application here is similar to the UNIFAC (UNIquac Functional group Activity Coefficients) model as parameterized by Peng et al. (2001). Peng et al. (2001) introduced an amendment of certain UNIFAC main group interaction parameters to improve simultaneously the model-measurement agreement of water activity for a series of aqueous dicarboxylic acids systems, including the straight-chain dicarboxylic acids from oxalic to glutaric acid as well as functionalized di- and tricarboxylic acids, such as tartaric acid, malic acid and citric acid. The amended parameter set is also applied in the AIOMFAC model. Recently, Marsh et al. (2017) studied the performance of the UNIFAC model in comparison to new measurements of pure and substituted dicarboxylic acids, sugars and amino acids. They report good agreement for the straight-chain dicarboxylic acids, but increasing deviations between model and measurements for alkyl-substituted dicarboxylic acids as the number of alkyl substitutions is increased. This is partially explained by the lack of experimental data for alkyl-substituted dicarboxylic acid systems during UNIFAC model parameter estimation work in the past. Acknowledging this weakness of UNIFAC-based models, we note that the methylsuccinic acid oxidation products of interest in this study are characterized by hydroxyl or ketone groups as additions/substitutions. Given the training of the Peng et al. (2001) UNIFAC parameters with mixtures containing malic acid and citric acid, it is likely that presence of such functional groups leads to a smaller systematic error on the hygroscopicity predictability of AIOMFAC compared to alkyl substituents. Therefore, the agreement between the AIOMFAC model prediction of water activity vs. water content of methylsuccinic acid and our measurements as well as those by Marsh et al. (2017) is likely not unique to this compound; rather, it is reasonable to expect a good (but not excellent) predictability of the hygroscopicity of methylsuccinic acid products after oxidation (and of their aqueous mixtures) at ~85 % RH.
We have added the following information in the revised manuscript.

Page 6 Line 27, "Recent measurements by Marsh et al. (2017) using the comparative kinetics technique applied in an electrodynamic balance (CK-EDB) also report hygroscopicity data for methylsuccinic acid (see **Fig. 1**), and the performance of the UNIFAC model in comparison to new measurements of pure and substituted dicarboxylic acids, sugars and amino acids."

Page 7 Line 8, "On the other hand, Marsh et al. (2017) reported good agreement between model and measurements for the straight-chain dicarboxylic acids, but increasing deviations for alkyl-substituted dicarboxylic acids as the number of alkyl substitutions is increased. The methylsuccinic acid oxidation products of interest in this study are characterized by hydroxyl or ketone groups. Peng et al. (2001) have introduced an amendment of certain UNIFAC main group interaction parameters to improve simultaneously the model-measurement agreement of water activity for a series of aqueous dicarboxylic acids systems, including the straight-chain dicarboxylic acids from oxalic to glutaric acid as well as functionalized di- and tricarboxylic acids, such as tartaric acid, malic acid and citric acid. The amended parameter set is also applied in the AIOMFAC model. Given the training of the Peng et al. (2001) UNIFAC parameters with mixtures containing malic acid and citric acid, it is likely that presence of such functional groups leads to a smaller systematic error on the hygroscopicity predictability of AIOMFAC compared to alkyl substituents. Therefore, given the uncertainties associated with both the CK-EDB data of Marsh et al. (2017) and the AOT data reported here, there is overall a good agreement between the experimental and modelled hygroscopicity data, and it is reasonable to expect a good predictability of the hygroscopicity of methylsuccinic acid and its oxidation products at 85 % RH."

*P6 L15-23 a long justification is given for possible differences between the CK-EDB and laser tweezer experiment. Since the two measurements agree with each other, within experimental errors, I'm not sure the justification is required?*

**Author Response:**
While the comparison of individual experimental data points with the data by Marsh et al. (2017) indicates agreement within stated measurement uncertainties (random error), where applicable, the overall comparison of the two data series as collections of points, describing hygroscopicity curves, indicates a small systematic difference. We would like to provide explanations for the difference between our experimental data and previously published data. This would help the reader to better understand these two measurements and techniques. We decided to keep the discussion in the revised manuscript.

*P7 Line 22. The error bars on the gamma value need to be acknowledged. I think the large error bars preclude a discussion of the gamma likely being greater than 1. For OH, a gamma of 1 is expected.*

**Author Response:**
We agree with the reviewer that in our experiment the value of gamma can be less or greater than one if we consider the error bars. We would also like to note that for the OH reaction with many organic compounds, the gamma measured by the decay of the parent compounds is found to be less than one. Although the gamma value of much greater than one supports the occurrence of

the secondary chemistry, the secondary chemistry could still occur when the gamma is less than one. As discussed in the text, in order to explain the large alcohol-to-ketone functionalization product ratio, secondary chemistry did likely occur, owing to the intermolecular hydrogen abstraction of an alkoxy radical.

*P11 Line 3 When the error in the gamma measurement is propagated through the model what is the outcome?*

**Author Response:**
We have re-run the model and included two scenarios using the upper and lower limit of the errors of gamma. The revised figures are included in the manuscript.

---

## Author Comment (AC2) · 20 Sep 2017

*The authors present experimental results on the OH oxidation of methyl succinic acid particles, which were generated from solution and maintained as aqueous particles at high humidity (RH=85%). Accompanying measurements of the water content of methyl succinic acid particles under varying humidity conditions are also presented. Experimental results for the oxidation rate of the methyl succinic acid and the hygroscopicity of methyl succinic acid are then used to model the changes in composition and size of methyl succinic acid particles as they are oxidized by OH. The model uses simplified chemistry (only two oxidation products and no secondary oxidation) with a sophisticated thermodynamic model (AIOMFAC) that predicts water content/particle size taking into account the changing activity coefficients of the methyl succinic acid, the two oxidation products, and water activity. The major conclusions are that oxidized methyl succinic acid particles increase in hygroscopicity and lose a notable amount of organic mass due to formation of volatile fragmentation products. The work presented is clearly described and presented fairly well. Understanding the oxidation of water soluble organic material, such as the di-acid presented here, is important in understanding the effects of water on particle oxidation processes. I recommend publication pending minor revisions.*

We would like to sincerely thank the reviewer for his/her thoughtful comments and suggestions. Please see our responses to reviewer's comments and suggestions below.

**General Comments**

*The authors must clarify the range of oxidation conditions. The high levels of oxidation are equivalent to a week in the atmosphere even for a moderate to high level of OH (2E6 mol/cm^3). A time axis for a given atmospheric OH level should be added to Figure 3.*

**Author Response:**
We have clarified that the high levels of oxidation used in this work are equivalent to 8.5 days in the atmosphere for a moderate to high level of OH concentration ($2 \times 10^6$ molecule cm$^{-3}$). This information is added to **Fig. 3** and the manuscript.

Page 5 Line 19, "The high levels of oxidation used in this work are equivalent to 8.5 days in the atmosphere for a moderate to high level of OH concentration ($2 \times 10^6$ molecule cm$^{-3}$)."

*A better sense of the mass balance during the experiments is needed, particularly for the volatilization. Some analysis of the amount of organic material lost to the gas phase must be presented. Using a simple mass balance and the assumed (or known) product hygroscopicities, how much fragmentation/volatilization is suggested given your experimental results? The simplified chemical mechanism is hard to reconcile with the results in Figure 3, because it would appear that the ratio of the products should always be (0.57/0.37). It seems like the predicted decrease in particle size would require a greater extent of fragmentation/volatilization. In any case the results in Figure 3 should be more clearly discussed in terms of the simplified chemical mechanism.*

**Author Response:**
We have analyzed the amount of organic material (i.e. fragmentation products) lost to the gas phase from our model simulations. The simulated particle-phase carbon mass normalized to its initial mass as a function of OH exposure is plotted in **Fig. 6** to investigate the extent of fragmentation and volatilization during oxidation. As shown in **Fig. 6**, about 7 % of the particle-phase carbon mass is lost at the maximum OH exposure. The relative abundance of major fragmentation product remaining in the particle phase is estimated to be 5 % as shown in **Fig. 3**. Volatilization and gas-phase oxidation of the fragmentation product is predicted to be significant as a greater fraction of the fragmentation products formed from oxidation is volatilized to the gas phase under our experimental conditions. This information is added in the revised manuscript.

Page 15 Line 15, "To quantify the amount of fragmentation products volatilized to the gas phase during oxidation, the normalized particle-phase carbon mass is plotted against OH exposure in **Fig. 6**. Model simulations show that about 7 % of the carbon mass is lost at the maximum OH exposure, while the relative abundance of particle-phase fragmentation product is about 5 %. Volatilization and gas-phase oxidation of the fragmentation product is predicted to be significant under the experimental conditions."

[Figure]

**Figure 6. Simulated normalized total carbon mass during heterogeneous OH oxidation of aqueous methylsuccinic acid droplets at 85 % RH.**

*The linearity of your mass spectrometry measurements should be shown. All the analysis presented assumes that the response of your mass spectrometer is linear over the wide range of MSA and product concentrations. A calibration using known amounts of MSA, and the linearity of response up to signal levels of 10ˆ9, should be shown.*

**Author Response:**
We agree with the reviewer's comment. However, the linearity of the methylsuccinic acid mass spectrometry measurement has not been tested when we performed the experiments about three years ago. In a separate study performed at the same time, we have measured the response of the mass spectrometer at different mass concentration of succinic acid particles using the same experimental setup. A quasi-linear response was observed over a range of aerosol mass concentration (see the graph below). As the methylsuccinic acid is structurally similar to succinic acid and very similar experimental conditions were applied in these two measurements, we assume that the response of the mass spectrometer is also linear for methylsuccinic acid.

[Figure]

**Specific Comments**

*Page 7 Line (10) It should be directly stated that this rate constant is an effective rate constant for OH radicals with aqueous methyl succinic acid particles.*

**Author Response:**
We have revised the sentence in the manuscript.

Page 8 Line 28, "the decay of methylsuccinic acid due to oxidation by OH exhibits an exponential behavior and can be fit with an exponential function to obtain an effective rate constant ($k$) for OH radicals with aqueous methylsuccinic acid particles:"

*It needs to be clear that the rate constant for OH + MSA was not directly measured in a single phase. 14 (5) "As the oxidation proceeds further (i.e. to the higher OH exposures), the formation of the fragmentation product becomes more significant " It is not clear how the model accounts for this. Do the alpha values change during the course of the oxidation? Is secondary oxidation of the functionalization and fragmentation products taken into account? It seemed that the model description specifically does not include the secondary oxidation.*

**Author Response:**
In the model, we assume that alpha values did not change during oxidation. Secondary oxidation of the functionalization and fragmentation products is not considered since the determined abundance of the second or higher generation reaction products is not significant. With the original sentence, we intended to mention that more fragmentation products are formed (in absolute, cumulative terms) as more methylsuccinic acid is oxidized; we did not intend to state that the fragmentation process become more favorable than the functionalization processes at higher oxidation stages. We have clarified this point and revised the sentence in the manuscript.

Page 15 Line 28, "As the oxidation proceeds further (i.e. to the higher OH exposures), more fragmentation product is formed in accumulative amount since more methylsuccinic acid is oxidized."

*14(12) "The largest deviation is observed at the maximum OH exposure. This could be explained by that for the particle composition (Fig. 3), the model-experiment discrepancy increases with increasing OH exposure, as discussed in the preceding section." This point is nearly lost in the awkward sentence construction. Please re-word such as: "The large deviation in particle size observed at the maximum OH exposure can be explained by the poorly predicted particle composition at high OH exposure (Fig. 3)."*

**Author Response:**
We have revised the sentence in the manuscript.

Page 16 Line 5, "The larger deviation in particle size observed at the maximum OH exposure can be explained by the poorly predicted particle composition at high OH exposure (**Fig. 3**)."

*14(30) "net hygroscopicity of the aerosols is slightly enhanced due to the formation of more oxidized functionalization products." Hygroscopicity is an intensive property, so the term "net" hygroscopicity lacks a clear meaning. In fact you present that the hygroscopicity of the particle organic content increases. In other words, the activity of water is further suppressed. The particles lose water due to loss of mass of soluble material. Please remove the term "net hygroscopicity" and re-word this sentence more similarly to Page 1 line 33.*

**Author Response:**
We have revised the sentence in the manuscript.

Page 16 Line 21, "Although the oxidized droplets can uptake more water than unreacted ones (relative to the organic content), the hygroscopicity of the aerosols is reduced as the number of water molecules is found to decrease at the entire OH exposure."